# UnicornNet: Unimodal Calibrated Ordinal Regression Neural Network

## Abstract

Ordinal regression is a supervised machine learning technique aimed at predicting the value of a discrete dependent variable with an ordered set of possible outcomes. Many of the algorithms that have been developed to address this issue rely on maximum likelihood for training. However, the standard maximum likelihood approach often fails to adequately capture the inherent order of classes, even though it tends to produce well-calibrated probabilities. Alternatively, some methods use Optimal Transport (OT) divergence as their training objective. Unlike maximum likelihood, OT accounts for the ordering of classes; however, in this manuscript, we show that it doesn't always yield well-calibrated probabilities. To overcome these limitations, we introduce UnicornNet, an approach inspired by the well-known Proportional Odds Model, which offers three key guarantees: (i) it ensures unimodal output probabilities, a valuable feature for many real-world applications; (ii) it employs OT loss during training to accurately capture the natural order of classes; (iii) it provides well-calibrated probability estimates through a post-training accuracy-preserving calibration step. Experimental results on seven real-world datasets demonstrate that UnicornNet consistently either outperforms or performs as well as recently proposed deep learning approaches for ordinal regression. It excels in both accuracy and probability calibration, while also guaranteeing output unimodality. The code will be publicly available upon acceptance.

## 1 Introduction

Ordinal regression is an area of supervised machine learning, where the goal is to predict the value of a discrete dependent variable, whose set of (symbolic) possible values is ordered. Despite often being overshadowed by more common tasks like classification and regression, ordinal regression covers a wide range of important applications, such as medical severity grading, credit rating, age estimation, and many more (De Vente et al., 2020; Wienholt et al., 2024; Niu et al., 2016; Kim & Ahn, 2012).

Ordinal regression is often treated as either a classification or a regression problem (for example, Kaggle's 2015 Diabetic Retinopathy competition[1] and Zha et al. (2023); Wu et al. (2023)) However, it lies between the two: Like classification, it predicts discrete labels, but unlike standard classifiers, it accounts for the inherent order between classes. Formally, ordinal regression assumes a label space $\mathcal{Y} = \{1, 2, \ldots, k\}$, endowed with a total order relation $\preceq$, such that $1 \preceq \ldots \preceq k$. A straightforward way to leverage the label order is to use a cost-sensitive loss function — one in which the penalty for misclassification increases strictly with the ordinal distance between the predicted and true labels. For example, predicting class "1" when the true class is "4" is considered more severe than predicting "3". Unlike regression, which operates over a continuous numeric range and is sensitive to monotonic transformations, ordinal regression treats labels as ordered but not numerically spaced. As a result, dedicated ordinal methods can outperform standard classifiers or regressors when the target variable is finite and ordered.

A fundamental ordinal regression model is the Proportional Odds Model (POM) (McCullagh, 1980), a generalized linear model similar in spirit to logistic regression, however the logits are defined for their cumulative probabilities. One possible limitation of POM, shared by several recent approaches to deep

---

[1] https://www.kaggle.com/c/diabetic-retinopathy-detection/

ordinal regression, is the common and often reasonable assumption that the predicted probability distribution should be unimodal. A $k$-level multinomial distribution is called *unimodal* if there exists $j \in \{1, \ldots, k\}$ such that $\mathbb{P}(Y = 1) \leq \ldots \leq \mathbb{P}(Y = j) \geq \ldots \geq \mathbb{P}(Y = k)$, where $Y \in \{1, \ldots, k\}$ is a random variable. Although there are domains in which unimodality is not necessarily a desirable property, such as tasks where the most common targets tend to fall at the extremes, in many other real-world domains it is a natural requirement, for example, when predicting a grade of a tumor, it may be counter-intuitive to trust a model prediction which says that a predicted tumor's grade is either "1" or "4", but not "2" or "3". However, unimodality is unfortunately not always fulfilled despite often being a desired characteristic. While this was identified by several recent works for deep ordinal regression (Gao et al., 2017; Diaz & Marathe, 2019; Liu et al., 2019a; 2020), unimodality is often encouraged (but not enforced) via soft targets. However, as we show in Section 2, using soft targets is suboptimal for achieving unimodality.

Another essential feature of an ordinal regression model is its ability to effectively capture the ordered relationship among classes within its training objective, while still reflecting the certainty of the model in its predictions. POM is typically trained via maximum likelihood, similar to several recently proposed deep ordinal regression approaches (Belharbi et al., 2019; Vargas et al., 2020; Fu et al., 2018; Beckham & Pal, 2017; Berg et al., 2020; Vishnu et al., 2019). We argue that maximum likelihood is not a suitable measure of quality in the context of ordinal regression, as it only considers the probability mass the model assigns to the true class, ignoring the remaining mass, and behaves as a non-cost-sensitive loss. This implicitly assumes that "all mistakes are equal", which, as discussed above, is not ideal for ordinal regression. However, a benefit of maximum likelihood is that it tends to yield well-calibrated probabilities. A well-calibrated model ensures that, for example, a 90% predicted probability corresponds to events that actually occur 90% of the time. However, many models, including recent deep ordinal regression approaches, struggle with this. These models often exhibit overconfidence or underconfidence, meaning their probability outputs are unreliable indicators of their true prediction certainty. Another commonly used loss function for ordinal regression tasks is Optimal Transport (OT) divergence (Hou et al., 2016; Beckham & Pal, 2017; Liu et al., 2019a). OT excels at capturing the inherent order between labels, potentially making it a better fit. However, as we explain in Section 3.3, OT might lead to peaked output distribution that lacks calibration.

In this manuscript, we therefore focus on two main contributions. First, We present UnicornNet, a novel approach for ordinal regression, based on deep learning machinery, which tackles the three issues pointed out above (i) it contains a mechanism to enforce *unimodality of the output distribution*, implemented via architectural design, (ii) it *effectively captures the ordered relationship among classes* using OT as a training objective (iii) it undergoes a post-training calibration process to output *well-calibrated probability estimates* that reflect the model's confidence in its predictions while still preserving the model's accuracy. Second, as discussed in Section 3.3, we identify a trade-off between certain requirements, noting that OT may prioritize peaked distributions over calibrated ones, which, to the best of our knowledge, was not pointed out in the literature in the context of deep ordinal regression. Importantly, this bonds the requirements together, as other methods that utilize OT as a training objective, end up being uncalibrated. We present experimental results on seven real-world image benchmark datasets which demonstrate that UnicornNet consistently performs on par with and often better than several recently proposed approaches for deep ordinal regression in terms of both prediction accuracy and probability calibration while having an unimodality guarantee.

## 2 Related Work

Being a traditional area of machine learning and statistics, there exists a large corpus of literature on ordinal regression. In this section, we focus on approaches based on recent deep-learning architectures. Several such approaches have been proposed in recent years. One common approach is to turn the ordinal regression problem into a multi-label classification problem (Fu et al., 2018; Liu et al., 2017; 2018b; Vishnu et al., 2019; Berg et al., 2020; Cheng et al., 2008; Li et al., 2021), where the task is framed as predicting a sequence of binary decisions indicating whether the target label exceeds each possible threshold. We argue that the multi-label approach has two major problematic aspects: first, the output probabilities are not always guaranteed to be consistent, in the sense of increasing cumulative distribution (i.e., we would like to predict $\mathbb{P}(y \leq 1) \leq \mathbb{P}(y \leq 2) \leq \ldots \leq \mathbb{P}(y \leq k)$. Second, even if the output probabilities are consistent, as is the case in Liu et al. (2018a); Shi et al. (2023); Cao et al. (2020), the predicted class probabilities are not necessarily

unimodal. This is the case in several recent works (Liu et al., 2019b; Vargas et al., 2020; Pan et al., 2018; Kook et al., 2020). Threshold-based methods are another widely-used family of approaches for ordinal regression, relying on a one-dimensional transformation of the input followed by discrete thresholding. Yamasaki & Tanaka (2024) has shown that such methods can struggle when the conditional distribution of the labels is non-unimodal or heteroscedastic, and that the choice of loss function plays a critical role in generalization performance. Also, contrastive representation learning has been applied to ordinal regression to capture label order in the latent space. Rank-N-Contrast (Zha et al., 2023) and SupReMix (Wu et al., 2023) improve performance by learning structured embeddings through ranking and mix-up strategies. While effective at modeling ordinal relations internally, they do not constrain the output layer and thus lack guarantees on unimodality, and probabilistic calibration, key properties for reliable ordinal regression.

One elegant mechanism to obtain unimodal output probabilities is to model the output conditional distribution as either the Poisson or the Binomial distributions (Beckham & Pal, 2017), which are both unimodal. In both cases, the model outputs a scalar ($\lambda$ in the case of the Poisson, $p$ in the case of the binomial) for each prediction, which is then mapped to a probability mass function that is used (after normalization) as the model output probabilities. Moreover, this method also learns a dataset-wide $\tau$ parameter which controls the shape of the output distribution. While being a convenient architectural-based solution for handling unimodality, this approach is inherently limited in its ability to express the level of uncertainty of the model's prediction. To see why, note that since a single parameter determines both the location of the mode and the decay of the probabilities, the model cannot output a highly flat or highly peaked probability vector, for example.

A different approach to unimodality has been to train the model with soft targets (Gao et al., 2017; Diaz & Marathe, 2019; Liu et al., 2019a; 2020), where the ground-truth label is replaced with a smoothed probability distribution over classes that reflects a pre-defined ordinal structure. For example, instead of using a one-hot encoding of the label, an exponentially decaying distribution is used. However, we argue that the utilization of soft targets suffers from two important disadvantages. First, unimodality is only encouraged, but not enforced. In Section 5 we will show cases where models trained with soft targets yield large amounts of non-unimodal predictions. Second, soft targets have a pre-defined decay pattern, which is determined a-priori and hence does not reflect any level of uncertainty with respect to the prediction. Therefore, they are equivalent (in the sense of a 1:1 map) to Dirac predictions (i.e., "one-hot"), and are devoid of any probabilistic insight whatsoever. As we show in this paper, our approach attends to both issues: we guarantee unimodal outputs, by design, and yield well-calibrated probabilities outputs that reflect the model's uncertainty. Other approaches for handling unimodality include Li et al. (2022); Cardoso et al. (2023), where unimodality is encouraged through a dedicated loss term (although not guaranteed), and Belharbi et al. (2019), which employs constrained optimization to achieve unimodality on the training data, but with no guarantees on the predictions on test data.

Several works use cross entropy as a training objective while using one-hot (or binary) targets (Belharbi et al., 2019; Vargas et al., 2020; Fu et al., 2018; Beckham & Pal, 2017; Berg et al., 2020; Vishnu et al., 2019; Cardoso et al., 2023; Yamasaki, 2022). Also, an accuracy-preserving calibration on a cross-entropy trained model was proposed in (Esaki et al., 2024). However, in the case of one-hot targets, the cross entropy term equals the negative logarithm of the probability assigned by the model to the true class, making it invariant to the distribution of the remaining probability mass and resulting in a non-cost sensitive loss. While reasonable in a standard classification setting, this ignores the order of the classes, making it less effective in ordinal regression settings that prioritize metrics sensitive to the ordering of classes, such as MAE, or other distance-based criteria that reflect the ordinal structure. Nonetheless, as cross-entropy is a proper-scoring rule, it tends to yield calibrated probability estimates (Lakshminarayanan et al., 2017). To address cross-entropy's limitation in handling the order of classes, some approaches use OT loss (Hou et al., 2016; Beckham & Pal, 2017; Liu et al., 2019a), which is a natural way to incorporate the order of the classes into the loss term. However, OT tends to favor peaked output distributions instead of calibrated ones (see Section 3.3). UnicornNet aims to benefit from both the ability to capture the order of classes using OT as a training objective, and the generation of calibrated probability estimates through accuracy-preserving calibration.

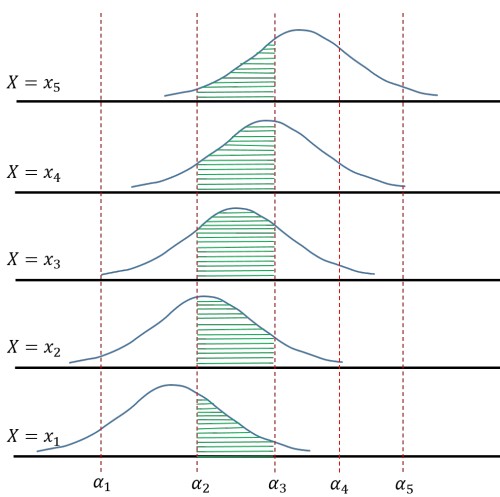

Figure 1: The proportional odds model. $\boldsymbol{x_i}$ is a realization of $\boldsymbol{X}$. The standard logistic density is shifted by $\boldsymbol{\beta}^T \boldsymbol{x_i}$. The thresholds $\boldsymbol{\alpha}_j$ define the bins which determine the probability predicted by the model to each class. For example, the green area defines the probability $\mathbb{P}(Y = 3|\boldsymbol{X} = \boldsymbol{x}; \boldsymbol{\alpha}, \boldsymbol{\beta})$.

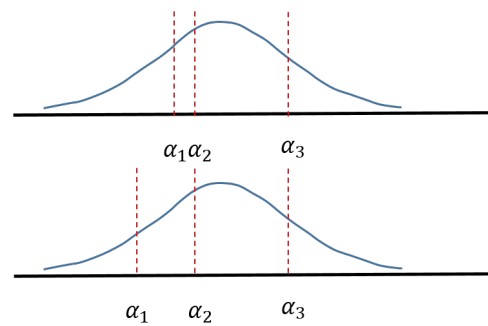

Figure 2: POM deficiencies. The plots illustrate two instances of POM, which highlight its two deficiencies: (1) POM does not always output unimodal probabilities: the first plot shows an example where the output probabilities are such that $\mathbb{P}(Y = 1|\boldsymbol{X} = \boldsymbol{x}; \boldsymbol{\alpha}, \boldsymbol{\beta}) > \mathbb{P}(Y = 2|\boldsymbol{X} = \boldsymbol{x}; \boldsymbol{\alpha}, \boldsymbol{\beta}) < \mathbb{P}(Y = 3|\boldsymbol{X} = \boldsymbol{x}; \boldsymbol{\alpha}, \boldsymbol{\beta}) > \mathbb{P}(Y = 4|\boldsymbol{X} = \boldsymbol{x}; \boldsymbol{\alpha}, \boldsymbol{\beta})$. (2) The likelihood function of POM is invariant to the way the predicted probability mass of the incorrect classes is assigned: if the correct class is 3, both instances have the same likelihood, even though in the second instance, the probability mass assigned to neighboring class 2 is larger.

In summary, to the best of our knowledge, no existing work has successfully met all three fundamental requirements for an effective ordinal regression model: (i) Ensuring unimodality in the output distribution, ideally through architectural design; (ii) Aligning the training objective function with the ordinal nature of the label space; (iii) Reflecting the model's uncertainty in the decay of the output probabilities, preferably with well-calibrated probabilities. These requirements led to the development of UnicornNet, presented in this paper.

## 3 Preliminaries

### 3.1 The Proportional Odds Model

Let $(\boldsymbol{X}, Y) \in \mathcal{X} \times \mathcal{Y}$ be random variables, having joint probability $\mathcal{P}_{XY}$, where $\mathcal{X} = \mathbb{R}^d$, $\mathcal{Y} = \{1, \ldots, k\}$, and $1, \ldots, k$ are considered as symbols. Let $\preceq$ be an order relation defined on $\mathcal{Y}$ such that $1 \preceq \ldots \preceq k$. The proportional odds model is parametrized by $\boldsymbol{\alpha} \in \mathbb{R}^{|\mathcal{Y}|-1}, \boldsymbol{\beta} \in \mathbb{R}^d$ and applies to data $\{(\boldsymbol{x_i}, y_i)\}_{i=1}^n$, sampled i.i.d. from $\mathcal{P}_{XY}$. Let $\epsilon$ be a logistic random variable (thus having a sigmoid cumulative distribution function $F(x) = \frac{1}{1+\exp(-x)}$). The entries of $\boldsymbol{\alpha}$ are used to define the cumulative conditional probabilities via

$$\mathbb{P}(Y \preceq j|\boldsymbol{X} = \boldsymbol{x}; \boldsymbol{\alpha}, \boldsymbol{\beta}) = \mathbb{P}(\boldsymbol{\beta}^T \boldsymbol{x} + \epsilon \leq \boldsymbol{\alpha}_j|\boldsymbol{X} = \boldsymbol{x}; \boldsymbol{\alpha}, \boldsymbol{\beta}) = F(\boldsymbol{\alpha}_j - \boldsymbol{\beta}^T \boldsymbol{x}) \tag{1}$$

Similarly to logistic regression, this yields linear log-odds (logits), however, defined with respect to cumulative terms

$$\gamma_j \equiv \log \frac{\mathbb{P}(Y \preceq j|\boldsymbol{X} = \boldsymbol{x})}{\mathbb{P}(Y \succ j|\boldsymbol{X} = \boldsymbol{x})} = \boldsymbol{\alpha}_j - \boldsymbol{\beta}^T \boldsymbol{x}$$

It is convenient to interpret equation 1 by viewing $\boldsymbol{\beta}^T \boldsymbol{x}$ as a factor that shifts the standard logistic density function, while the $\alpha_j$ terms are thresholds, with respect to which the cumulative probabilities are defined. This is depicted in Figure 1.

Let $(\boldsymbol{x}, y)$ be a realization of $(\boldsymbol{X}, Y)$. The likelihood assigned by the model to $(\boldsymbol{x}, y)$ is

$$\mathbb{P}(Y = y | \boldsymbol{X} = \boldsymbol{x}; \boldsymbol{\alpha}, \boldsymbol{\beta}) = F(\alpha_y - \boldsymbol{\beta}^T \boldsymbol{x}) - F(\boldsymbol{\alpha}_{y-1} - \boldsymbol{\beta}^T \boldsymbol{x}), \tag{2}$$

considering $\boldsymbol{\alpha}_0 = -\infty$ and $\boldsymbol{\alpha}_k = \infty$. The model is typically trained in a standard fashion by maximizing the log-likelihood function on the training data.

Despite its popularity, POM has two key limitations: First, the model's output probabilities are not necessarily unimodal (see Figure 2). Second, the likelihood function in equation 2 depends only on the probability the model assigns to the correct class $y$ and is invariant to the way the remaining probability mass is assigned by the model. This ignores the order on the label set, and hence does not use important information that might be used to improve prediction quality, as depicted in Figure 2. This is also true of ordinal likelihood proposed in Chu et al. (2005). Additionally, when the target labels are one-hot encoded, the cross-entropy loss—being equivalent to the negative log-likelihood—depends solely on the predicted probability of the true class. As a result, it is invariant to how the remaining probability mass is distributed among the incorrect classes. In Section 4 we will show how UnicornNet overcomes these two limitations of POM.

## 3.2 Probability Calibration

Probability Calibration refers to the alignment between the predicted probabilities generated by a classification model and the true, empirical probabilities observed in the data. For instance, it is anticipated that when a classification model assigns a probability of 0.8 to class $i$ for a certain sample, approximately 80% of those samples would indeed belong to class $i$ based on their ground truth labels. Formally, according to Wang (2023), the concept of calibration can be defined as:

$$\mathbb{P}(Y = i | p_i(\boldsymbol{X}) = q) = q, \quad \forall q \in [0, 1], i \in [k] \tag{3}$$

Here, $k$ denotes the number of classes, $\boldsymbol{X}$ and $Y$ are random variables that represent the model input and the ground truth class, respectively, and $p_i(\boldsymbol{X})$ is a random variable that signifies the model's output probability for class $i$ given the random variable $\boldsymbol{X}$.

Denote $\boldsymbol{e_Y} \in \mathbb{R}^k$ as the one-hot encoding random vector of $Y$, then, we can express an equivalent definition to equation 3: A classification model is calibrated if for all $\boldsymbol{x}$ $\mathbb{E}_{Y|\boldsymbol{X}=\boldsymbol{x}}[\boldsymbol{e_Y} | p(\boldsymbol{x})] = p(\boldsymbol{x})$. In practice, $p(x)$ is a one-to-one function, hence we can refine the definition further: $\mathbb{E}_{Y|X=x}[\boldsymbol{e_Y} | \boldsymbol{x}] = p(\boldsymbol{x})$.

To achieve calibration, minimizing the squared $L_2$ norm on the difference between the two sides of this equation is desirable. However, given the unknown true conditional distribution of $Y$ given $X = x$, we can only approximate $\mathbb{E}_{Y|\boldsymbol{X}=\boldsymbol{x}}[\boldsymbol{e_Y} | \boldsymbol{x}]$. One approach is to use a Monte Carlo approximation with one sample, yielding the approximation $\boldsymbol{e_y}$, i.e., minimize the following:

$$BS(\boldsymbol{e_y}, p(\boldsymbol{x})) = \|\boldsymbol{e_y} - p(\boldsymbol{x})\|_2^2 \tag{4}$$

This term is known as the Brier Score (BS) (Brier, 1950). BS is a well-known Proper Scoring Rule (Dawid & Musio, 2014), which evaluates the accuracy of probabilistic predictions. Proper scoring rules are optimized when the probabilistic forecast matches the true probability distribution of outcomes. Therefore, we will use BS as a training objective to calibrate the model predictions.

**Accuracy-Preserving Calibration**   An accuracy-preserving calibration method is a technique that adjusts the probability outputs of a pre-trained model to improve their calibration, without affecting the model's accuracy. A popular method is Temperature Scaling (TS) (Guo et al., 2017), which learns a single temperature parameter that is used to rescale the model's logits before applying the softmax activation, adjusting the confidence predictions. Adaptive Temperature Scaling (ATS) (Balanya et al., 2024) extends TS such that instead of a single temperature parameter, it learns a mapping $\boldsymbol{x} \mapsto T(\boldsymbol{x})$ that adaptively scales the logits based on the input $x$. As part of UnicornNet we define an accuracy-preserving calibration method with similarity to TS and ATS which is trained via Brier Score (Section 4.3).

### 3.3 Optimal Transport

Let $M$ be a finite metric space with moving cost metric $c(\boldsymbol{x}, \boldsymbol{y})$ between elements $\boldsymbol{x}, \boldsymbol{y} \in M$, and let $\boldsymbol{p}, \boldsymbol{q}$ be probability mass functions on $M$. The optimal transport (OT) or 1-Wasserstein distance between $\boldsymbol{q}$ and $\boldsymbol{p}$ is defined as:

$$OT(\boldsymbol{p}, \boldsymbol{q}) = \inf_{\gamma \in \Gamma} \int_{M \times M} c(\boldsymbol{x}, \boldsymbol{y}) d\gamma(\boldsymbol{x}, \boldsymbol{y}), \tag{5}$$

Where $\Gamma$ is the set of joint probabilities on $M \times M$ with marginals $\boldsymbol{q}$ and $\boldsymbol{p}$, and $c$ specifies moving costs between elements of $M$. This computes the optimal way to transport $\boldsymbol{q}$ into $\boldsymbol{p}$. When $\boldsymbol{q}$ is a Dirac (one-hot), OT simplifies to:

$$OT(\boldsymbol{p}, \boldsymbol{q}) = \sum_{i=1}^{k} \boldsymbol{p}_i c(i, j), \tag{6}$$

Where $j$ is the correct class and $k$ is the number of classes. With model outputs $\boldsymbol{p}$ and one-hot target $\boldsymbol{q}$, this loss is differentiable w.r.t. $\boldsymbol{p}$. The cost metric $c$ can encode domain knowledge. For ordered classes $M = 1, ..., k$, a natural cost is $c(i, j) = |i - j|^m$ for some $m \geq 1$.

Another theoretical motivation for incorporating OT in ordinal regression arises from Proposition 5.2 in Frogner et al. (2015). Let $\mathcal{H}$ be a hypothesis class of functions $h_\theta : \mathcal{X} \rightarrow \Delta^k$, where $\Delta^k$ denotes the probability simplex over $k$ ordinal labels. Let $h_{\hat{\theta}}$ be the OT empirical risk minimizer. For an input $\boldsymbol{x}$, let $h_\theta(\boldsymbol{x}) \in \Delta^k$ be the predicted distribution, and $\boldsymbol{e_y}$ the one-hot vector corresponding to the true label $y \in \{1, \ldots, k\}$. We define $\hat{y}_{\hat{\theta}}(\boldsymbol{x}) := \arg\max_j h_{\hat{\theta}}(\boldsymbol{x})_j$ to be the predicted class. Then:

**Proposition 3.1.** *(Frogner et al., 2015) For the cost function $c(i, j) = |i - j|$ and any $\delta > 0$, with probability at least $1 - \delta$ it holds that*

$$\mathbb{E}_{\boldsymbol{x}, y} \left[ |\hat{y}_{\hat{\theta}}(\boldsymbol{x}) - y| \right] \leq \inf_{h_\theta \in \mathcal{H}} k \cdot \mathbb{E} \left[ OT(h_\theta(\boldsymbol{x}), \boldsymbol{e_y}) \right] + C$$

Here, $C$ refers to a constant term that depends on $k, \delta$, the number of samples, the metric and properties of the hypothesis space, such as its Rademacher complexity. The proposition implies that minimizing the OT loss leads to lower mean absolute error (MAE) by encouraging predictions that respect the ordinal structure of the labels, making it a well-suited loss function for ordinal regression tasks.

However, one drawback of OT as a loss function, is its tendency to prioritize peaked distributions over the actual probabilities. For instance, let $\boldsymbol{X}, Y$ be random variables represents the input and the label, and let $\boldsymbol{x}$ be the realisation of $\boldsymbol{X}$. We also denote $\boldsymbol{e_Y}$ to be a random variable which is the one-hot encoded vector representation of $Y$. Consider the conditional probabilities $\mathbb{P}(Y = 1 | \boldsymbol{X} = \boldsymbol{x}) = 0.25, \mathbb{P}(Y = 2 | \boldsymbol{X} = \boldsymbol{x}) = 0.5, \mathbb{P}(Y = 3 | \boldsymbol{X} = \boldsymbol{x}) = 0.25$. Now, suppose there are two model outputs: $\boldsymbol{p} = [0.25, 0.5, 0.25]$ and $\boldsymbol{q} = [0, 1, 0]$. To achieve calibration, the model should ideally output $\boldsymbol{p}$ as it equals the true conditional distribution of $Y$ given $\boldsymbol{X} = \boldsymbol{x}$. However, it is notable that for the cost metric $c(i, j) = |i - j|$, we find that $OT(\boldsymbol{p}, \boldsymbol{e_y}) = 1$ with a combined probability of $0.25 + 0.25 = 0.5$, and $OT(\boldsymbol{p}, \boldsymbol{e_y}) = 0.5$ with a probability of $0.5$. Consequently, $\mathbb{E}_{Y|\boldsymbol{X}=\boldsymbol{x}}[OT(\boldsymbol{p}, \boldsymbol{e_Y})] = 0.75$, and similarly, $\mathbb{E}_{Y|\boldsymbol{X}=\boldsymbol{x}}[OT(\boldsymbol{q}, \boldsymbol{e_Y})] = 0.5$. This observation indicates that OT tends to favor peaked distributions over calibrated ones. UnicornNet addresses this challenge, as described in Section 4.3.

## 4 UnicornNet

In this section, we describe our novel mechanism for unimodal calibrated ordinal regression neuralnetwork, UnicornNet an approach for architectural-based generation of unimodal output probability distributions, as well as accuracy-preserving probability calibration.

### 4.1 Rational

Achieving unimodality directly via architectural design has a major advantage since the output probabilities are guaranteed to be unimodal for every input instance, as is also the case for the mechanism proposed

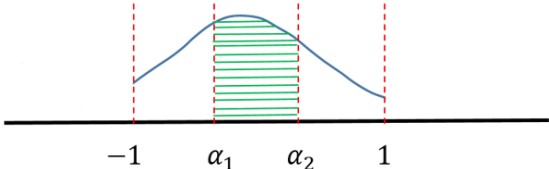

Figure 3: Generation of unimodal output probabilities for $k = 3$ classes. An input $\boldsymbol{x}$ is mapped to a $(\mu, \sigma)$ pair, which defines a truncated normal distribution $\mathcal{N}_{\text{trunc}}(\mu, \sigma^2, -1, 1)$ over the real line. The output probabilities are proportional to the mass in the bins, which are of equal length. The green area equals to probability $p_2(\boldsymbol{x})$, corresponding to $\mathbb{P}(Y = 2 | \boldsymbol{X} = \boldsymbol{x})$.

in Beckham & Pal (2017). However, UnicornNet employs the truncated normal distribution, depending on two parameters $(\mu, \sigma)$, which influence the location of the mode and the decay of the probability mass. This adds flexibility to the shape of the output probability vector, compared to the mechanism in Beckham & Pal (2017) where a single parameter determines both the mode and the decay. For determining the mode of the distribution, a map $x \mapsto \mu$ is learned via OT using equation 6. However, as mentioned in Section 3.3, OT tends to favor peaked distributions over calibrated ones. To address this issue, we adopt an accuracy-preserving calibration strategy where, subsequent to learning a map $\boldsymbol{x} \mapsto \mu$, the model further learns a map $\boldsymbol{x} \mapsto \sigma$ by optimizing the Brier Score using equation 4, while maintaining the previously learned map $\boldsymbol{x} \mapsto \mu$ fixed.

## 4.2 Unimodal Output Probabilities Generation

Inspired by POM, we utilize thresholds to define bins, so that the total mass inside each bin is the output probability of the corresponding class. However, we observe that the lack of unimodality of POM can be fixed by letting the bins be of equal length and remain fixed during training.

Therefore, instead of learning the thresholds, during training two maps $\boldsymbol{x} \mapsto \mu, \boldsymbol{x} \mapsto \sigma$ are learned, where $\mu$ is a location parameter, and $\sigma$ is a scale parameter. Both define a truncated normal distribution $\mathcal{N}_{\text{trunc}}(\mu, \sigma^2, -1, 1)$ (a normal distribution that has the same density as the normal density on [-1,1], normalized to have a unit integral, and is zero outside this interval), from which the output probabilities are derived.

Formally, we divide the range $[-1, 1]$ to $k$ equal bins similarly to da Costa et al. (2008), where $k$ is the number of classes, defined by $-1 = \alpha_0, \alpha_1, ..., \alpha_k = 1$, so that $\alpha_i - \alpha_{i-1} = \frac{2}{k}$. The probabilities are given by

$$p_i(\boldsymbol{x}) = \mathbb{P}(Y = i | \boldsymbol{X} = \boldsymbol{x}; \mu, \sigma) = F_{\mu(\boldsymbol{x}), \sigma(\boldsymbol{x})}(\alpha_i) - F_{\mu(\boldsymbol{x}), \sigma(\boldsymbol{x})}(\alpha_{i-1}), \tag{7}$$

where $F_{\mu, \sigma}(\cdot)$ is the $\mathcal{N}_{\text{trunc}}(\mu, \sigma^2, -1, 1)$ cumulative distribution function, and note that $\mu, \sigma$ are in fact functions of the input instance $\boldsymbol{x}$.

To compensate for the fact that the probability-generating mechanism depends on fewer parameters than POM (2 for the former, $d + k - 1$ for the latter), the maps $\boldsymbol{x} \mapsto \mu, \boldsymbol{x} \mapsto \sigma$ are expressed via two deep neural networks (DNNs) which share a common backbone model, hence can represent a complex nonlinear relation. Our proposed mechanism for the generation of unimodal output probabilities is depicted in Figure 3.

The following lemma, proved in Appendix A , establishes that the model output probabilities are indeed unimodal.

**Lemma 4.1.** *Let $\boldsymbol{x} \in \mathbb{R}^d$ be an input to the model, which is mapped to $\mu = \mu(\boldsymbol{x}), \sigma = \sigma(\boldsymbol{x})$. Let $p_1, \ldots p_k$ be the model output probabilities, generated via equation 7. Then $p_1, \ldots p_k$ define a unimodal multinomial random variable.*

## 4.3 Accuracy-Preserving Calibration in UnicornNet

UnicornNet introduces a re-training procedure for $\boldsymbol{x} \mapsto \sigma$ mapping while preserving the already trained mapping $\boldsymbol{x} \mapsto \mu$. The learning of $\boldsymbol{x} \mapsto \sigma$ is done by minimizing the Brier Score loss in equation 4, which

---

**Algorithm 1** Optimal Transport-based Training

---

**Input:** Training data $\mathcal{D} = \{(\boldsymbol{x_i}, y_i)\}_{i=1}^N$, number of classes $k$, batch size $b$, number of epochs $T$
**Output:** Parameters $\theta_\mu$, $\theta_\sigma$ of $\mu(\cdot)$ and $\sigma(\cdot)$, respectively, and backbone parameters $\phi$

1: Initialize parameters $\theta_\mu$, $\theta_\sigma$, $\phi$
2: **for** epoch $t = 1$ to $T$ **do**
3:    **for** each batch $\mathcal{B} \subseteq \mathcal{D}$ of size $b$ **do**
4:       Compute $\mu_{\phi,\theta_\mu}(\boldsymbol{x_i})$ and $\sigma_{\phi,\theta_\sigma}(\boldsymbol{x_i})$ for all $\boldsymbol{x_i} \in \mathcal{B}$ and calculate $p(\boldsymbol{x_i})$ using Eq.7
5:       Compute the loss
         $\mathcal{L}_{\mathrm{OT}}(\mathcal{B}) = \frac{1}{b} \sum_{(\boldsymbol{x_i}, y_i) \in \mathcal{B}} \mathrm{OT}(p(\boldsymbol{x_i}), y_i)$ using Eq.6
6:       Update $\theta_\mu$, $\theta_\sigma$, $\phi$ via Gradient Descent
7:    **end for**
8: **end for**
9: Return $\theta_\mu$, $\theta_\sigma$, $\phi$

---

ensures well-calibrated probabilities and preserves the model accuracy. Since the mapping $\boldsymbol{x} \mapsto \sigma$ controls the decay of the generated probabilities and the mapping $\boldsymbol{x} \mapsto \mu$ which controls the mode, remains unchanged, the accuracy and MAE is not affected, although the probability output distribution is indeed modified, resulting in an accuracy-preserving calibration We state it in the following lemma (proved in Appendix B).

**Lemma 4.2.** *Let $\boldsymbol{x} \in \mathbb{R}^d$ be an input to the model, mapped to $\mu = \mu(\boldsymbol{x})$, $\sigma_1 = \sigma_1(\boldsymbol{x})$. Let $\sigma_2 = \sigma_2(\boldsymbol{x})$ be a re-trained mapping via minimization of loss using equation 4. Let $p_1^{\sigma_1}, \ldots, p_k^{\sigma_1}$ and $p_1^{\sigma_2}, \ldots, p_k^{\sigma_2}$ be the model output probabilities generated using $(\mu, \sigma_1)$ and $(\mu, \sigma_2)$, respectively, via equation 7. Then:*

$$\mathrm{argmax}_{1 \le i \le k}\, p_i^{\sigma_1} = \mathrm{argmax}_{1 \le i \le k}\, p_i^{\sigma_2}.$$

Intuitively, to preserve accuracy, the distribution should be unimodal and symmetric around its mode. This motivates our choice of the truncated normal distribution.

### 4.4 Training Procedure

To summarize, UnicornNet's training process consists of two distinct phases, as detailed in Algorithms 1, 2. In the first phase, described in Algorithm 1, the parameters $\theta_\mu$ and $\theta_\sigma$ of $\mu(\cdot)$ and $\sigma(\cdot)$, respectively, along with the backbone parameters $\phi$, are jointly optimized using Gradient Descent to minimize the OT loss defined in Eq. 6.

In the second phase, outlined in Algorithm 2, the parameters $\theta_\mu$ and $\phi$ are kept fixed, while $\theta_\sigma$ is further optimized via Gradient Descent to minimize the BS defined in Eq. 4.

---

**Algorithm 2** Accuracy Preserving Calibration

---

**Input:** Training data $\mathcal{D} = \{(\boldsymbol{x_i}, y_i)\}_{i=1}^N$, number of classes $k$, batch size $b$, number of epochs $T$, parameters
    $\theta_\mu$ of $\mu(\cdot)$ and backbone parameters $\phi$
**Output:** Parameters $\theta_\sigma$ of $\sigma(\cdot)$

1: **for** epoch $t = 1$ to $T$ **do**
2:    **for** each batch $\mathcal{B} \subseteq \mathcal{D}$ of size $b$ **do**
3:       Compute $\mu_{\phi,\theta_\mu}(\boldsymbol{x_i})$ and $\sigma_{\phi,\theta_\sigma}(\boldsymbol{x_i})$ for all $\boldsymbol{x_i} \in \mathcal{B}$ and calculate $p(\boldsymbol{x_i})$ using Eq. 7
4:       Compute the loss
         $\mathcal{L}_{\mathrm{BS}}(\mathcal{B}) = \frac{1}{b} \sum_{(\boldsymbol{x_i}, y_i) \in \mathcal{B}} \mathrm{BS}(p(\boldsymbol{x_i}), y_i)$ using Eq. 4
5:       Update **only** $\theta_\sigma$ via Gradient Descent
6:    **end for**
7: **end for**
8: Return $\theta_\sigma$

---

# 5 Experimental Results

## 5.1 Datasets

We evaluate UnicornNet on seven real-world benchmark image datasets, involving various ordinal regression tasks: age-detection (Adience Eidinger et al. (2014), FG-Net Fu et al. (2014), AAF Cheng et al. (2019)), facial beauty prediction (SCUT-FBP5500 Liang et al. (2018)), bio-medical image classification (Retina-MNIST Yang et al. (2023)), image aesthetics estimation (EVA Kang et al. (2020)), and tabular (Fireman [2] dataset). A more detailed description of the datasets appears in Appendix D . Some examples from the Adience and Retina-MNIST datasets are shown in Figure 4.

## 5.2 Benchmark

We compare UnicornNet to seven recently presented approaches for deep ordinal regression, with unimodal output probabilities and to a deep learning approach of POM:

- DLDL (Gao et al., 2017), an approach utilizing soft labels, generated using squared exponentially decaying distributions, trained using Kullback-Leibler divergence minimization (equivalent to cross-entropy minimization).

- SORD (Diaz & Marathe, 2019), an approach utilizing soft labels, generated using linear exponentially decaying distributions, trained using Kullback-Leibler divergence minimization.

- Beckham & Pal (2017), an architectural-based approach in which unimodal output probabilities are generated using the binomial distribution (single-learned parameter), trained using optimal transport loss

- Liu et al. (2019a), an approach utilizing soft labels, created as a mixture of Dirac, uniform, and linear exponentially decaying distributions, trained using optimal transport loss

- POM (McCullagh, 1980), a variant of the POM, incorporating a deep learning model with a POM layer integrated on top, trained using cross-entropy loss

- UnimodalNet (Cardoso et al., 2023), a non-parametric architectural-based approach for unimodality, trained using cross-entropy loss.

- CORN[3] (Shi et al., 2023), a method ensuring rank-consistent ordinal regression by modeling conditional probabilities over binary classification tasks, trained on conditional data subsets and using the probability chain rule, while avoiding restrictive weight-sharing constraints.

- ORCU (Kim et al., 2025), a loss function that jointly promotes unimodality and well-calibrated confidence estimates via soft ordinal encoding and a ordinal-aware regularization term.

## 5.3 Experimental Setup

To ensure a fair comparison, we implemented all methods using the same image transformations, backbone CNN, and training procedures. This setup guarantees that the only differences among the methods lie in their output layer architectures and loss functions. All models were trained using the Adam optimizer with default parameters. The selected hyperparameters can be found in Table 9.

Each experiment was repeated five times to account for variability due to random initialization and data splits. For all datasets except Adience and Fireman, the train-validation-test splits were randomly generated and held consistent across methods. In the case of the Adience dataset, we followed the original evaluation protocol and repeated experiments across the five predefined splits provided by the dataset creators[4], in case

---

[2]https://github.com/gagolews/ordinal_regression_data
[3]https://github.com/Raschka-research-group/corn-ordinal-neuralnet/blob/main/model-code/simple-scripts/mlp_corn.py
[4]https://github.com/GilLevi/AgeGenderDeepLearning/tree/master/Folds/train_val_txt_files_per_fold

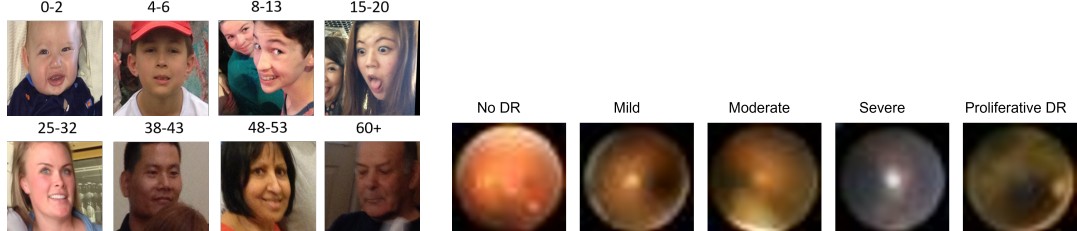

Figure 4: Left: Examples from the Adience dataset. The age category is indicated above each image. Right: Examples from the Retina mnist dataset. Diabetic Retinopathy classes are indicated above.

of Fireman, we used the balanced version as presented in Shi et al. (2023), and the same train, validation and test sets.

The means and standard deviations reported in Table 1 reflect performance over these five repetitions. For the accuracy-preserving calibration phase, the number of training epochs and learned parameters varied by dataset detailed in Table 10. For reproducibility, the supplementary material contains code reproducing the results reported in this section.

## 5.4 Evaluation Metrics

We report several commonly-used evaluation metrics for ordinal regression tasks: MAE (Baccianella et al., 2009), One-Off Accuracy (OOA) (ELKarazle et al., 2022), Spearman correlation (Spearman, 1961), Quadratic Weighted Kappa (QWK) (Cohen, 1968), as well as the percentage of test examples with unimodal predicted output probabilities.

In addition, following the discussion in section 3.2, we evaluate the model's probability calibration using the well known Expected Calibration Error (ECE) (Guo et al., 2017). ECE quantifies the discrepancy between predicted probabilities and actual outcomes, measuring how well the predicted probabilities of a model align with the true likelihood of the predicted events. To compute ECE, the $[0, 1]$ interval is first divided into a set of $b$ equal-length bins. ECE is defined as:

$$ECE = \sum_{i=1}^{b} \frac{|B_i|}{|X|} |\text{Acc}(B_i) - \text{Conf}(B_i)|,$$

where $b$ is the number of probability bins (in all of our experiments $b = 10$), $\text{Acc}(B_i)$ is the empirical accuracy in bin $B_i$ and $\text{Conf}(B_i)$ is the average predicted probability in bin $B_i$. An ECE of 0 indicates perfect calibration, while higher values signify miscalibration, with the model being over-confident or under-confident.

## 5.5 Results on Real World Datasets

Table 1 shows the test results of each method on the seven benchmark datasets. As can be seen, Unicorn-Net performs at least on par and often better than the compared baselines, in a fairly consistent manner, across the various datasets and evaluation metrics, specifically with respect to the MAE, where UnicornNet outperforms on all the datasets except Fireman with neglible difference. In addition, observe that only UnicornNet, UnimodalNet, ORCU and Beckham & Pal (2017) output consistently unimodal probabilities, while the other baselines, trained using soft targets, do not always output unimodal probabilities. Yet, unlike UnimodalNet and Beckham & Pal (2017), which tend to produce poorly calibrated probability estimates, UnicornNet consistently produce well calibrated probability estimates, where on most of the datasets, UnicornNet outperforms all baselines, while on the other datasets, the difference from ORCU is negligible. To sum up the results, UnicornNet produces well-calibrated, unimodal probability distribution while it does not compoermise on the accuracy-based metrics. Additional experiments conducted on other dataset and models are presented in Appendix C.

Table 1: Performance of various methods on real world datasets, in a mean ± std format

| Dataset | Method | MAE ↓ | OOA ↑ | Spearman ↑ | QWK ↑ | % Unimodal ↑ | ECE ($b=10$) ↓ |
|---|---|---|---|---|---|---|---|
| Adience | Beckham and Pal | .50 ± .06 | .93 ± .01 | **.90 ± .01** | **.91 ± .02** | **1 ± 0** | .23 ± .05 |
| | Liu et al. | .47 ± .05 | .94 ± .01 | .89 ± .02 | .90 ± .02 | .47 ± .04 | .21 ± .03 |
| | DLDL | .49 ± .06 | .93 ± .01 | .88 ± .02 | .89 ± .03 | .62 ± .08 | .42 ± .04 |
| | SORD | .47 ± .06 | .94 ± .01 | .89 ± .02 | .90 ± .02 | .99 ± .003 | .14 ± .03 |
| | POM | .48 ± .06 | .94 ± .01 | .89 ± .02 | **.91 ± .03** | .81 ± .04 | .32 ± .04 |
| | UnimodalNet | .49 ± .06 | .93 ± .01 | .88 ± .02 | .90 ± .03 | **1 ± 0** | .34 ± .04 |
| | CORN | .52 ± .07 | N/A | N/A | N/A | N/A | N/A |
| | ORCU | **.45 ± .06** | **.95 ± .01** | **.90 ± .02** | **.91 ± .03** | **1 ± 0** | **.06 ± .01** |
| | UnicornNet | .46 ± .05 | **.95 ± .01** | **.90 ± .02** | **.91 ± .02** | **1 ± 0** | .07 ± .03 |
| Retina MNIST | Beckham and Pal | .80 ± .02 | .79 ± .01 | .58 ± .02 | .55 ± .02 | **1 ± 0** | .17 ± .01 |
| | Liu et al. | .68 ± .02 | **.82 ± .01** | **.61 ± .02** | .58 ± .02 | .72 ± .03 | .27 ± .01 |
| | DLDL | .72 ± .02 | .81 ± .01 | .59 ± .02 | .58 ± .01 | .98 ± .01 | .29 ± .01 |
| | SORD | .75 ± .02 | .78 ± .01 | .58 ± .007 | .57 ± .007 | .87 ± .03 | .11 ± .01 |
| | POM | .83 ± .02 | .76 ± .01 | .53 ± .02 | .53 ± .02 | .43 ± .03 | .17 ± .01 |
| | UnimodalNet | .74 ± .02 | .80 ± .01 | .59 ± .01 | .58 ± .01 | **1 ± 0** | .14 ± .01 |
| | CORN | .76 ± .02 | N/A | N/A | N/A | N/A | N/A |
| | UnicornNet | **.67 ± .009** | **.82 ± .007** | **.61 ± .01** | **.59 ± .01** | 1 ± 0 | **.06 ± .01** |
| FG-NET | Beckham and Pal | .44 ± .06 | .95 ± .03 | .80 ± .04 | .79 ± .04 | **1 ± 0** | .16 ± .02 |
| | Liu et al. | .32 ± .08 | .96 ± .02 | .86 ± .03 | .84 ± .04 | .01 ± .01 | .08 ± .02 |
| | DLDL | .40 ± .09 | .96 ± .02 | .81 ± .03 | .81 ± .03 | .12 ± .04 | .49 ± .06 |
| | SORD | .36 ± .08 | .97 ± .02 | .84 ± .05 | .82 ± .04 | .99 ± .01 | .26 ± .05 |
| | POM | .33 ± .06 | .97 ± .02 | .84 ± .04 | .85 ± .04 | .41 ± .06 | .21 ± .03 |
| | UnimodalNet | .36 ± .07 | .97 ± .01 | .84 ± .02 | .84 ± .03 | **1 ± 0** | .25 ± .04 |
| | CORN | .34 ± .07 | N/A | N/A | N/A | N/A | N/A |
| | ORCU | .33 ± .07 | .98 ± .03 | .85 ± .03 | .86 ± .04 | **1 ± 0** | .11 ± .02 |
| | UnicornNet | **.31 ± .08** | **.99 ± .02** | **.87 ± .03** | **.88 ± .01** | **1 ± 0** | **.07 ± .02** |
| AAF | Beckham and Pal | .61 ± .13 | .91 ± .05 | .81 ± .03 | .80 ± .05 | **1 ± 0** | .16 ± .04 |
| | Liu et al. | .43 ± .01 | **.97 ± .005** | .82 ± .01 | **.85 ± .01** | .7 ± .17 | .22 ± .01 |
| | DLDL | .54 ± .02 | .95 ± .01 | .77 ± .01 | .77 ± .02 | .95 ± .02 | .32 ± .02 |
| | SORD | .44 ± .02 | .96 ± .01 | .82 ± .01 | .84 ± .01 | **1 ± 0** | .14 ± .02 |
| | POM | .45 ± .01 | .96 ± .01 | .80 ± .01 | .83 ± .01 | .48 ± .07 | .28 ± .02 |
| | UnimodalNet | .45 ± .01 | .96 ± .01 | .80 ± .01 | .83 ± .01 | **1 ± 0** | .30 ± .01 |
| | CORN | .46 ± .01 | N/A | N/A | N/A | N/A | N/A |
| | ORCU | .44 ± .02 | .96 ± .00 | .82 ± .01 | .84 ± .01 | **1 ± 0** | .04 ± .02 |
| | UnicornNet | **.42 ± .01** | **.97 ± .005** | **.83 ± .01** | **.85 ± .005** | **1 ± 0** | **.03 ± .01** |
| SCUT-FBP5500 | Beckham and Pal | .62 ± .12 | .92 ± .04 | .84 ± .02 | .81 ± .05 | **1 ± 0** | .19 ± .08 |
| | Liu et al. | .54 ± .03 | .95 ± .01 | .83 ± .01 | .82 ± .01 | .18 ± .14 | .27 ± .02 |
| | DLDL | .68 ± .05 | .90 ± .01 | .79 ± .02 | .73 ± .02 | .86 ± .08 | .28 ± .04 |
| | SORD | .57 ± .03 | .94 ± .01 | .83 ± .01 | .81 ± .01 | .99 ± .005 | .09 ± .02 |
| | POM | .53 ± .03 | .95 ± .01 | .83 ± .01 | .83 ± .01 | .28 ± .04 | .33 ± .03 |
| | UnimodalNet | .52 ± .02 | .95 ± .01 | .82 ± .01 | .83 ± .01 | **1 ± 0** | .35 ± .01 |
| | CORN | .58 ± .03 | N/A | N/A | N/A | N/A | N/A |
| | ORCU | .52 ± .04 | .95 ± .01 | .83 ± .01 | .83 ± .02 | **1 ± 0** | **.05 ± .01** |
| | UnicornNet | **.47 ± .02** | **.96 ± .01** | **.85 ± .01** | **.85 ± .01** | **1 ± 0** | .06 ± .02 |
| EVA | Beckham and Pal | .61 ± .03 | .93 ± .01 | **.60 ± .01** | **.60 ± .02** | **1 ± 0** | .11 ± .01 |
| | Liu et al. | .65 ± .02 | .91 ± .02 | .53 ± .01 | .52 ± .02 | .72 ± .05 | .31 ± .01 |
| | DLDL | .66 ± .03 | .91 ± .01 | .53 ± .02 | .52 ± .02 | .98 ± .01 | .20 ± .01 |
| | SORD | .60 ± .02 | .93 ± .01 | .58 ± .02 | .57 ± .02 | **1 ± 0** | **.07 ± .02** |
| | POM | .65 ± .03 | .91 ± .01 | .53 ± .02 | .53 ± .03 | .89 ± .02 | .45 ± .02 |
| | UnimodalNet | .66 ± .03 | .91 ± .01 | .52 ± .02 | .52 ± .02 | **1 ± 0** | .45 ± .02 |
| | CORN | .67 ± .02 | N/A | N/A | N/A | N/A | N/A |
| | ORCU | .62 ± .02 | .93 ± .01 | .55 ± .02 | .55 ± .02 | **1 ± 0** | .09 ± .02 |
| | UnicornNet | **.57 ± .01** | **.94 ± .005** | **.6 ± .01** | .58 ± .01 | **1 ± 0** | .08 ± .02 |
| Fireman | Beckham and Pal | .84 ± .02 | .81 ± .01 | **.97 ± .003** | **.97 ± .002** | 1 ± 0 | .09 ± .01 |
| | Liu et al. | 1.36 ± .05 | .62 ± .04 | .92 ± .001 | .92 ± .001 | .01 ± .01 | .45 ± .03 |
| | SORD | 1.01 ± .01 | .76 ± .001 | .95 ± .003 | .95 ± .004 | .93 ± .03 | .17 ± .00 |
| | POM | .77 ± .002 | .85 ± .001 | **.97 ± .001** | **.97 ± .001** | .01 ± .005 | .08 ± .001 |
| | UnimodalNet | **.74 ± .01** | **.86 ± .003** | **.97 ± .002** | **.97 ± .00** | 1 ± 0 | **.06 ± .005** |
| | CORN | .76 ± .01 | N/A | N/A | N/A | N/A | N/A |
| | ORCU | .84 ± .01 | .81 ± .01 | .97 ± .002 | .97 ± .003 | 1 ± 0 | .22 ± .003 |
| | UnicornNet | .75 ± .04 | .84 ± .02 | **.97 ± .003** | **.97 ± .002** | 1 ± 0 | **.06 ± .01** |

## 5.6 Computational Cost

In this section, the computational cost associated with both the backbone training and the calibration phase is analyzed. The calibration stage introduces minimal overhead in comparison to the primary training

phase. Specifically, it involves the optimization of only the scale head parameters $\theta_\sigma$, while the backbone model remains fixed throughout. Since $\theta_\sigma$ constitutes a parameter set that is several orders of magnitude smaller than that of the backbone, and no gradient computation or backpropagation is conducted through the backbone, the additional training cost incurred by the calibration phase is negligible.

To empirically validate the efficiency of the calibration phase, Table 2 presents the wall-clock training durations for both phases across three datasets. The results demonstrate that Phase 2 consistently requires substantially less time than Phase 1.

Table 2: Wall-clock training time comparison between Phase 1 (OT-based traning) and Phase 2 (accuracy preserving calibration).

| Dataset | Phase | Time (HH:mm) |
|---------|-------|--------------|
| EVA | Phase 1 | 04:03 |
| | Phase 2 | **00:47** |
| AAF | Phase 1 | 04:44 |
| | Phase 2 | **00:37** |
| FGNET | Phase 1 | 02:14 |
| | Phase 2 | **00:35** |

## 5.7 Hyperparameter Sensitivity

Overall, no unusual sensitivity was observed during training. Among the hyperparameters, the cost metric used in the OT loss plays the most critical role. This metric encodes prior knowledge about the ordinal relationship between classes and directly influences the magnitude of the gradient updates—larger cost values for distant class pairs amplify the penalization of large errors.

Empirical evidence suggests that selecting a cost metric with linear structure (i.e., $m = 1$) yields superior performance across multiple evaluation metrics. To further investigate the effect of this choice, an additional experiment was conducted on the FG-NET dataset, comparing the performance of two models with different cost metric hyperparameters: $m = 1$ and $m = 2$. The results, summarized in Table 3, demonstrate that a quadratic cost ($m = 2$) leads to degraded performance, confirming the importance of selecting an appropriate cost structure for the OT loss.

Table 3: Sensitivity of the cost metric parameter $m$ on performance metrics (FG-NET dataset).

| Method | MAE ↓ | OOA ↑ | Spearman ↑ | QWK ↑ | % Unimodal ↑ | ECE ($b$=10) ↓ |
|--------|-------|-------|------------|-------|--------------|----------------|
| UnicornNet ($m = 1$) | **.31 ± .08** | **.99 ± .02** | **.87 ± .03** | **.88 ± .01** | 1 ± 0 | **.07 ± .02** |
| UnicornNet ($m = 2$) | .35 ± .07 | .98 ± .01 | .85 ± .03 | .85 ± .02 | 1 ± 0 | .12 ± .03 |

## 5.8 Ablation Study

### 5.8.1 Calibration

In this section, we analyzed the impact of the calibration phase introduced in Section 4.3 on the ECE values of UnicornNet. Table 4 shows the ECE with and without the calibration phase on the Adience, EVA, AAF, and Retina MNIST datasets. Without calibration, the ECE is higher, indicating that the model's outputs are not well-calibrated due to the OT properties discussed in Section 3.3. This highlights the importance of incorporating the calibration step to accurately reflect the model's confidence in its predictions.

### 5.8.2 Optimal Transport

We further analyzed the impact of the OT loss on the MAE by conducting an experiment on the RetinaM-NIST dataset. Specifically, we compared two variants of UnicornNet the first trained with cross-entropy loss and the second trained with OT loss. The results, summarized in Table 5, show that OT outperforms cross-entropy in terms of MAE. This outcome aligns with expectations, as cross-entropy does not account for the inherent order of the classes, making OT a more appropriate choice for ordinal regression tasks.

Table 4: The effect of calibration phase on the ECE.

| Dataset | Calibration | ECE ($b = 10$) ↓ |
|---|---|---|
| Retina MNIST | yes | **.06 ± .01** |
| | no | .38 ± .007 |
| Adience | yes | **.07 ± .03** |
| | no | .18 ± .03 |
| AAF | yes | **.03 ± .01** |
| | no | .28 ± .01 |
| EVA | yes | **.08 ± .02** |
| | no | .32 ± .02 |

Table 5: Comparison of OT and Cross-Entropy Loss on RetinaM-NIST.

| Loss | MAE ↓ |
|---|---|
| Cross-entropy | .87 ± .01 |
| Optimal Transport | **.67 ± .01** |

## 6 Conclusion

In this manuscript, we identify several issues with current deep ordinal regression methods, including the potential for OT to result in poor probability calibration. We therefore presented UnicornNet, an approach for deep ordinal regression, inspired by the proportional odds model. UnicornNet utilizes an architectural mechanism for the generation of unimodal output probabilities, trained using OT objective and calibrated using an accuracy-preserving calibration process which encourages uncertainty awareness. We demonstrated that while performing on par with and often better than other recently proposed approaches for ordinal regression, the presented method enjoys the benefits of *guaranteed* unimodal and well-calibrated output probabilities.

## Broader Impact Statement

Our work focuses on ordinal regression models with applications in sensitive domains such as medical diagnosis and credit scoring. While UnicornNet improves both prediction accuracy and probability calibration, it is crucial to recognize that any residual miscalibration could lead to adverse outcomes, such as inappropriate treatment decisions or biased risk assessments. Future deployments should include thorough fairness and calibration audits, especially across demographic subgroups, to mitigate unintended consequences.

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

# A  Proof of Lemma 4.1

*Proof.* Let $p_i, p_{i+1}$ be the output probabilities of two adjacent classes, and let $-1 = \alpha_0, \ldots, \alpha_k = 1$ be the thresholds. We will show that (i) if $\mu \leq \alpha_{i-1}$ then $p_i \geq p_{i+1}$. Symmetrical argument will then imply that if $\mu \geq \alpha_{i+1}$ then $p_{i+1} \geq p_i$ (ii) if $\mu \in (\alpha_{i-1}, \alpha_i)$ then $p_i > p_{i+1}$, whenever the latter exists. Similarly, this would imply that $p_i > p_{i-1}$. Together, (i) and (ii) will prove the statement of the lemma.

Denote by $f$ the density of the $\mathcal{N}_{\text{trunc}}(\mu, \sigma^2, -1, 1)$ distribution. To prove (i) observe that $p_i > \frac{2f(\alpha_i)}{k} > p_{i+1}$.

To prove (ii), divide the $i$'th bin to two sub-bins $B_{i,1}, B_{i,2}$, of lengths $a = \mu - \alpha_{i-1}$ and $b = \alpha_i - \mu$, respectively. Similarly, divide the $(i+1)$th bin to two bins $B_{i+1,1}, B_{i+1,2}$ length $b$ and $a$, respectively. Then from (i)

$$\int_{B_{i,2}} f(x)dx > \int_{B_{i+1,1}} f(x)dx. \tag{8}$$

In addition, observe that

$$\begin{aligned}
\int_{B_{i,1}} f(x)dx &= \int_0^a f(\mu + x)dx \\
&> \int_0^a f(\mu + 2b + x)dx \\
&= \int_{B_{i+1,2}} f(x)dx.
\end{aligned} \tag{9}$$

Adding up equation 8 and equation 9, we obtain $p_i > p_{i+1}$. Apart from just being unimodal, we also proved that if $\mu \in (\alpha_{i-1}, \alpha_i)$ then class $i$ is the predicted class by the model.

$\square$

# B  Proof of Lemma 4.2

*Proof.* Let $-1 = \alpha_0 < \alpha_1 < \ldots < \alpha_k = 1$ be the thresholds. In the proof of Lemma 4.1 (Appendix A), we demonstrated that for any $i \in \{1, \ldots, k\}$, if $\mu \in (\alpha_{i-1}, \alpha_i)$, then class $i$ is the predicted class by the model.

Given that the two sets of parameters of the truncated normal distribution, $(\mu, \sigma_1^2)$ and $(\mu, \sigma_2^2)$ share the same mean $\mu$ and the thresholds $\alpha_0, \ldots, \alpha_k$ are constant, the predicted class $i$ remains the same for both sets of parameters regardless of the variances $\sigma_1^2$ and $\sigma_2^2$.

$\square$

# C  Additional Experiments

### C.1  Comparing UnicornNet against CORN on another dataset

In Table 6 we provide the MAE results of UnicornNet trained on the AFAD dataset additionally to the results from Shi et al. (2023) for CE-NN, OR-NN, CORAL and CORN. The results for these baselines were borrowed from Shi et al. (2023). We train UnicornNet with the same backbone as in Shi et al. (2023), Resnet-34. The dataset is balanced, and the train, validation, and test splits are the same as in Shi et al. (2023). We train UnicornNet 5 times with different random intialization seeds. Our method ourperforms the baselines on both datasets.

### C.2  Comparing UnicornNet against simple constraint-unimodal baseline

We conduct an additional experiment to compare our method with a simple baseline classifier trained using cross-entropy loss and a unimodality constraint. In this baseline, the model predicts a K-dimensional probability vector for each sample, and the unimodality constraint is enforced by minimizing pairwise distances

Table 6: Test MAE of methods trained on the AFAD dataset

| Method | MAE |
|--------|-----|
| CE-NN | $3.28 \pm 0.04$ |
| OR-NN | $2.85 \pm 0.03$ |
| CORAL | $2.99 \pm 0.03$ |
| CORN | $2.81 \pm 0.02$ |
| UnicornNet | $\mathbf{2.645 \pm 0.02}$ |

between the predicted probability values. Specifically, the distances are set to be negative for probabilities at indices lower than the target label and positive for indices higher than the target label. Further implementation details are available in the project repository.

The results, shown in Table 7, indicate that the baseline achieves near-unimodal distributions ( 80% unimodality) on the Fireman dataset but does not guarantee perfect unimodality. Additionally, the Mean Absolute Error (MAE) of this baseline is higher compared to UnicornNet. Notably, when applied to the AFAD dataset with the same training parameters, the baseline fails to produce unimodal predictions, highlighting its limitations in maintaining unimodality across different datasets.

Table 7: Performance Comparison on Balanced Datasets (AFAD and Fireman)

| Method | AFAD (balanced) | | Fireman (balanced) | |
|--------|-----------------|---|--------------------|---|
| | MAE $\downarrow$ | Unimodality $\uparrow$ | MAE $\downarrow$ | Unimodality $\uparrow$ |
| UnicornNet | $2.64 \pm 0.02$ | $1.0 \pm 0.0$ | $0.755 \pm 0.003$ | $1.0 \pm 0.0$ |
| Classifier with Unimodality | $2.846 \pm 0.04$ | $0.11 \pm 0.04$ | $0.808 \pm 0.003$ | $0.865 \pm 0.05$ |

## D  Datasets

Tabel 8 contains information on the benchmark datasets used for our experiments.

**Adience**: During the training the images are resized to (256,256). Additionally, random crop of size 224 and random horizontal flip are applied as augmentations.

**FG-Net**: We partitioned the dataset to 8 classes, corresponding to decades. Augmentations are the same as in the Adience experiment.

**RetinaMNIST** dataset has 5 classes, and we apply random affine, horizontal and vertical flips as augmentations during the training. The size of the images is (28,28) as provided by the dataset contributors. The train/test splits are proved by the contributors and were used as-is.

**SCUT-FBP5500** dataset contains 5500 face images beautifully ranked from 1 to 5 continuously. we partition the data into 8 classes in accordance with the rank. Augmentations are the same as in the Adience experiment.

**EVA (Explainable Visual Aesthetics)** dataset contains 5101 images aesthetically ranked from 0 to 10 by multiple voters. We calculate the average score for each image and partition the data into 5 classes in accordance with the average score. Augmentations are the same as in the Adience experiment.

**AAF (All-Age-Faces)** dataset is already pre-processed and contains 13,322 face images (mostly Asian), distributed across all ages (from 2 to 80). We partitioned the dataset into 6 classes. Augmentations are the same as in the Adience experiment.

**AFAD (The Asian Face data)**[5] dataset (Niu et al., 2016) contains 165,501 faces in the age range of 15-40 years. No additional preprocessing was applied to this dataset since the faces were already centered.

---

[5]https://github.com/afad-dataset/tarball

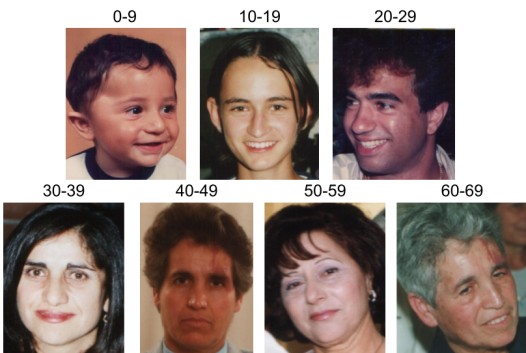

Figure 5: Examples from the FG-Net dataset. Age classes are indicated above each image.

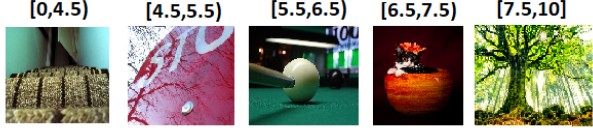

Figure 6: Examples from the EVA dataset. Aesthetics classes are indicated above each image.

Following Shi et al. (2023), we use a balanced version of the AFAD dataest[6] with 13 age labels in the age range of 18-30 years, resulting in total 60K samples.

**Fireman** [7] dataset is a tabular dataset that contains 40,768 instances, 10 numeric features, and an ordinal response variable with 16 categories. Just like Shi et al. (2023), we use a balanced version of this dataset[8] consisting of 2,543 instances per class and 40,688 from the 16 ordinal classes in total. The train, validation, and test splits are the same as in Shi et al. (2023)

Figures 5, 6, 7, 8 show examples from the FG-Net, EVA, AAF and SCUT-FBP5500 datasets, respectively.

## E Technical Details

Table 9 presents the technical specifications of the experiments conducted on the real-world benchmark datasets used in this study. Table 10 provides the number of learned parameters and training epochs used during the calibration phase.

## F Computation Details

We implement our model in Pytorch and run experiments on a Linux server with NVIDIA GeForce GTX 1080 Ti, A100 80GB PCIe GPUs and Intel(R) Core(TM) i7-8700 CPU 3.20GHz CPU.

---

[6]https://github.com/Raschka-research-group/corn-ordinal-neuralnet/tree/main/datasets/afad
[7]https://github.com/gagolews/ordinal_regression_data
[8]https://github.com/Raschka-research-group/corn-ordinal-neuralnet/tree/main/datasets/firemen

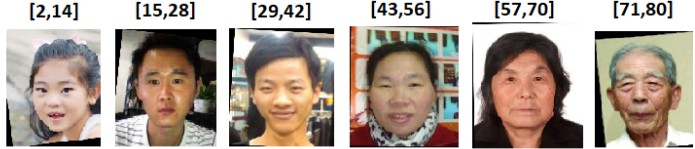

Figure 7: Examples from the AAF dataset. Age classes are indicated above each image.

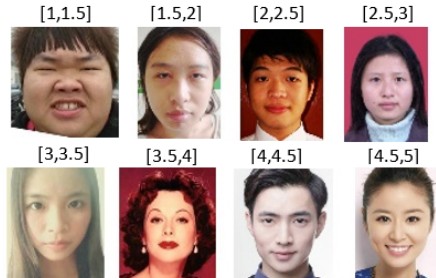

Figure 8: Examples from the SCUT-FBP5500 dataset. Beauty classes are indicated above each image.

Table 8: Benchmark datasets characteristics

| Dataset | Task | Train Images | Val Images | Test Images | Classes |
|---|---|---|---|---|---|
| Adience | age estimation | | pre-defined splits | | 8 |
| FG-Net | age estimation | 802 | 100 | 100 | 8 |
| RetinaMNIST | DR classification | 1080 | 120 | 400 | 5 |
| AAF | age estimation | 9058 | 1599 | 2665 | 6 |
| EVA | aesthetics estimation | 3684 | 651 | 766 | 6 |
| SCUT-FBP5500 | facial beauty prediction | 4250 | 350 | 900 | 8 |
| Fireman | tabular | 30922 | 1628 | 8138 | 16 |

Table 9: Technical details of the experiments

| Dataset | Backbone | Epochs | Batch Size | Initial LR | Decay LR After (epochs) | Weight Decay |
|---|---|---|---|---|---|---|
| Adience | ResNet-101 | 100 | 64 | $10^{-4}$ | 40 | $10^{-5}$ |
| FG-Net | ResNet-18 | 100 | 32 | $10^{-4}$ | 40 | $10^{-4}$ |
| RetinaMNIST | ResNet-18 | 100 | 16 | $10^{-4}$ | 80, 90 | $10^{-4}$ |
| AAF | ResNet-18 | 100 | 64 | $10^{-4}$ | - | $10^{-3}$ |
| EVA | ResNet-18 | 145 | 64 | $10^{-4}$ | - | $10^{-5}$ |
| SCUT-FBP5500 | ResNet-18 | 100 | 64 | $10^{-4}$ | - | $10^{-3}$ |
| Fireman | MLP(2 layers of 300) | 200 | 64 | $10^{-3}$ | 150,190 | $10^{-3}$ |

Table 10: Post-hoc calibration technical details

| Dataset | Epochs | Learned Parameters |
|---|---|---|
| Adience | 100 | 250 |
| FG-Net | 100 | 250 |
| RetinaMNIST | 100 | 1000 |
| AAF | 100 | 150 |
| EVA | 100 | 50 |
| SCUT-FBP5500 | 100 | 100 |
| Fireman | 30 | 150 |

