# OpenReview forum: "UNICORNN: Unimodal Calibrated Ordinal Regression Neural Network"
_TMLR — Withdrawn by Authors_

### Review · Reviewer_j8ZT · 2025-05-15

**Summary Of Contributions:**

This study tackles ordinal regression problem, which is a supervised learning problem aimed at predicting the value of a discrete dependent variable with an ordered set of possible outcomes.

The authors argue that minimization of optimal transport divergence accounts for the ordering of classes unlike the maximum likelihood method, but yields ill-calibrated probability prediction. Also, the authors recommend to use an unimodal likelihood model that can only represent unimodal conditional probability distribution, considering nature of data in ordinal regression problem.

Therefore, this study developes an ordinal regression method UNICORNN, which is consisted of three main components: The first is an unimodal likelihood model, which is inspired by the proportional odds model and uses a combination of truncated Gaussian cumulative distribution function and equal-length bins. The second is the optimal transport divergence to learn a model. The third is post-training accuracy-preserving calibration.

The authors argue that UNICORNN enjoys the benefits of guaranteed unimodal and well-calibrated output probabilities.

**Audience:**

Yes

**Claims And Evidence:**

No

**Requested Changes:**

My list of proposed adjustments to the submission are following C1-8.
Especially, C1-7 are all "critical to securing my recommendation for acceptance" because they relate to the correctness/appropriateness of the discussion and the handling of previous studies.
I cannot give any more specific instructions on what modifications should be made to make the paper acceptable.
These are the minimum requirements.

C1.
Change the name "UNICORNN" (related to W1).

C2.
Define "the inherent order of classes" clearly (related to W2).

C3.
Reconsider how to handle cost-sensitive learning (related to W3, 5).

C4.
Rewrite discussion of previous works carefully and more in detail (related to W3-8, 13, 14).

C5.
Rewrite mathematical discussion carefully (related to W5, 8, 18).

C6.
Cite references appropriately (related to W4, 16).

C7.
Reconsider experimental design (related to W3, 10, 14, 15).

C8.
Improve writing (related to W9, 11, 12, 16, 17, 19).

**Strengths And Weaknesses:**

Strengths

S1.
The proposed method of constructing a unimodal likelihood model using a truncated Gaussian cumulative distribution function and equal-length bins (Eq.7) may be novel.

S2.
Theoretical guarantees (Lemmas 4.1 and 4.2) are correct.

___

Weakness (*W~ is critical)

*W1.
It is better to avoid "UNICORNN", since there is a previous work titled with "Unicornn":

-Rusch, T. Konstantin, and Siddhartha Mishra. "Unicornn: A recurrent model for learning very long time dependencies." International Conference on Machine Learning. PMLR, 2021.

*W2.
What is "the inherent order of classes" (in Line 5 of Abstract)?
Does it stand for "unimodality of the underlying conditional probability distribution", or for others?
In either case, it is unclear in the current writing what "the inherent order of classes" stands for.
Authors should define it mathematically if possible.
The authors write "OT accounts for the ordering of classes" in Abstract, but I cannot understand this claim partly because "the ordering of classes" is not defined in the paper.

*W3.
I am concerned that the authors do not properly distinguish between "cost-sensitive learning" and "ordinal regression".
The sentence in Lines 12-16 of Section 1 is misleading.
When we want to maximize accuracy in ordinal regression task, all mistakes are equal.
I believe that the essence of ordinal regression is simply the ordinal relation behind the data, that cost-sensitive tasks are often considered in an ordinal regression framework, but ordinal regression framework also considers accuracy maximization.
Please clarify the relationship between "cost-sensitive learning" and "ordinal regression" or "the inherent order of classes" (see W2) following authors' idea.

Also, I worry that the authors are not aware of a standard treatment of cost-sensitive learning.
For example, when we learn logistic regression model $(p_{\\theta,1}(x),\\ldots,p_{\\theta,k}(x))$ (model of the conditional probablity distribution $(\\Pr(Y=1\mid X=x),\\ldots,\\Pr(Y=k\mid X=x))$) by maximum likelihood model, it is better to predict a label for $x$ by $\\arg\\min_y \\sum_{l=1}^k p_{\\theta,l}(x) |y-l|$ for minimization of MAE (cost-sensitive task).
However, authors' program code uses argmax for label prediction:
"preds = torch.argmax(preds, dim=-1)" of "class MAE(Metric)".
See $f_\\ell$ in P.5 of the previous work R1 or Bayes decision theory (see $\\pi^*(x)$ in https://vincent-maladiere.github.io/proba-ml/decision-theory/bayesian-decision-theory or search some textbooks).
In cost-sensitive learning, it is standard to change either the learning objective function or the decision function.
This study appears to have adopted the former strategy, but did not adequately address the latter strategy.
Perform experiments with the latter strategy as well.

Moreover, I feel that the discussion of cost-sensitive learning and the accuracy-preserving calibration do not go well together.
Cite additionally relevant reference studies if exists.

*W4.
The authors write "Another commonly used loss function for ordinal regression tasks is Optimal Transport (OT) divergence." in P.2.
Please cite previous ordinal regression works that used OT.
The current writing of Section 3.3 is insufficient.
I have theoretical concerns, such as whether the use of OT undermines the consistency of estimators of conditional probabilities, because of no reference and no discussion in the paper.
For example, does the use of OT (6) with $c(i,j)=|i-j|$ yield a Bayes optimal classifier for the criterion MAE?

*W5.
The authors write "We argue that maximum likelihood is a sub-optimal measure of quality for ordinal regression setup, as it only considers the probability mass the model assigns to the true class, ignoring the remaining mass" in P.2 as a criticism for maximum likelihood estimation.
This ("it only considers ...") is an error.
Consider the example where the dataset includes $(x_1,y_1)=(1,1), (x_2,y_2)=(1,1), (x_3,y_3)=(1,2)$ and $\\arg\\max_y \\Pr(Y=y\mid X=1)=1$ (I think this is what the authors call true class). There, maximum likelihood estimation considers $\\Pr(Y=2\mid X=1)$ as well.
Also, I cannot agree this claim under the knowledge of cost-sensitive learning (see W2).

*W6.
The authors write "first, the output probabilities are not always guaranteed to be consistent, in the sense of increasing cumulative distribution" in P.3 as a criticism for multi-label approach.
This is an error.
Consider the example where one models $\\Pr(Y\\le y\mid X=x)$ by $1/(1+e^{-g_y(x)})$.
We can ensure $1/(1+e^{-g_1(x)})\\le\\cdots\\le1/(1+e^{-g_{k-1}(x)})$ by implementing $g_1(x)=f_1(x)$ and $g_y(x)=g_{y-1}(x)+\\{f_y(x)\\}^2$ for $y=2,\\ldots,k-1$ with other models $f_1,\\ldots,f_{k-1}$.
See the previous works R1 or R2, or others.
This is no longer an issue of multi-label approach.

*W7.
The authors write "While being a convenient architectural-based solution for handling unimodality, this approach is inherently limited in its ability to express the level of uncertainty of the model’s prediction" in P.3 as a criticism for Beckham & Pal (2017).
However, this issue is completely resolved in (VS-SL model of) the previous work R1:
VS-SL model can represent any unimodal conditional probability distribution.

W8.
The authors write "However, we observe that the lack of unimodality of POM can be fixed by letting the bins be of equal length and remain fixed during training" at head of Section 4.2.
This is correct when using truncated Gaussian cumulative distribution function, but not generally correct (for general function $F$).
See Theorem 3 of the previous work R2.

*W9.
The authors can write more details about experimental settings since TMLR has no page limit.

W10.
Five trials in the experiments are small.
Increase the number of trials if possible.

W11.
Some sentences are difficult for me to understand (language issue):

・"One potential source of sub-optimality of POM and of several recently-proposed approaches for deep ordinal regression, is the often-reasonable requirement that a probabilistic model for ordinal regression will output unimodal probabilities" in P.2.

・"In addition, it is important to mention that as the cross-entropy term is essentially equivalent to the model’s negative log-likelihood function, this invariance to the partition of the remaining mass over the incorrect classes is common to all models trained via cross-entropy minimization, as long as the target labels are one-hot."

W12.
Do not use the word "suboptimal" (or "optimal") loosely.
For example, the authors write "We argue that maximum likelihood is a sub-optimal measure of quality for ordinal regression setup" in P.2.
Optimality is a (mathematical) concept that is defined with respect to some criterion, and a sentence containing that word sounds like a mathematical assertion.
However, in the above example, it is unclear with respect to what criterion it is suboptimal.
Choice another word, or rewrite the sentence more precise.

*W13.
Section 2 is not explanatory enough.
Readers can't understand the content at all without reading references.
For example, it says that the previous works (Gao et al., 2017; Diaz & Marathe, 2019; Liu et al., 2019a; 2020) use soft targets, but what kind of soft targets are they?

W14.
The authors should emphasize S1, but they have not provided sufficient comparison/discussion of the proposed unimodal likelihood model with other similar methods.
See Section 5.1 of the previous work R1 (proportional-odds variants).
It is difficult to calculate the cumulative distribution function of a truncated Gaussian distribution;
this makes it difficult to learn the location model $\\mu(x)$ and the scale model $\\sigma(x)$ simultaneously, and hence the proposed method requires post-calibration (two-stage learning of $\\mu$ and $\\sigma$);
(Beckham & Pal, 2017) and R1 methods do not have these difficulties, and the need for calibration even seems like a drawback of the proposed method.
Also, I am curious to see what would happen if we did a post-calibration to (Beckham & Pal, 2017) in the experiments.
These are just one example, and it needs a more detailed discussion of properties of the proposed unimodal likelihood model.

*W15.
Ablation studies are inadequate.
I understood that UNICORNN is consisted of three main components:
The first is an unimodal likelihood model, uses a combination of truncated Gaussian cumulative distribution function and equal-length bins.
The second is the optimal transport divergence to learn a model.
The third is post-training accuracy-preserving calibration.
This paper provide ablation studies regarding the third component only.
It needs ablation studies regarding the first and second components as well:
Compare a proposed unimodal likelihood model and other unimodal likelihood models in (Beckham & Pal, 2017) and the previous works R1 while keeping the training objective function the same, and compare OT minimization and maximum likelihood method (since the paper does not describe this comparison or references).
Otherwise, readers won't know which components are working properly.

W16.
Insufficient description of (or citation about) technical terms:

・I do not know One-Off Accuracy (OOA) in Section 5.3. It needs mathematical description or citation. The same is true for Mean Absolute Error (MAE), Spearman correlation, and Quadratic Weighted Kappa (QWK).

・The authors write "In addition, following the discussion in section 3.2, we evaluate the model’s probability calibration using the well known Expected Calibration Error (ECE)" in Section 5.3. It needs citation to show "well known".

W17.
Please unify significant digits of values in Tables.

W18.
The mathematical description is wrong or not sophisticated:

・$\\mathcal{Z}=\\beta^TX+\\epsilon$ (above Eq.1) -> $Z=\\beta^TX+\\epsilon$

・Introduction of $Z$ (above Eq.1) is unnecessary.

・It is natural that the most left-hand side term of Eq.1 means a conditional probability, but center term of Eq.1 means a joint probability. Therefore, the first equality of Eq.1 is incorrect.

・Introduction of $L$ (in Eq.2) is unnecessary.

・$\\mathbb{P}(\\cdot)$ in Eq.1, but $\\mathbb{P}[\\cdot]$ in Eq.3. These bracket formats are not unified.

・$Y$ is not defined (before Line 9 in P.2).

・It is difficult to distinguish between scalar and vector objects. I recommend use \bm{} or \\mathbf{} for a vector object.

・Eq.2 or Eq.7 can be represent in 1 line.

・"given the unknown real distribution $Y|X=x$" is loose: $Y|X=x$ is not a distribution.

・$\mathbb{P}(Y = 3)$ is a marginal probability. Wouldn't it be more appropriate to write the conditional probability $\mathbb{P}(Y = 3\mid X=x)$ here? Throughout the paper, the distinction between marginal probability and conditional probability is unclear.

・Is the definition (Eq.3) of calibration standard? Cite some reference. This is different from (1) of the following paper:

-Wang, C. (2023). Calibration in deep learning: A survey of the state-of-the-art. arXiv preprint arXiv:2308.01222.

W19.
There are many writing errors:

・"1" (in P.1) -> ``1'' (in Latex)

・Table D.1 -> ?

・i.i.d from (in Line 4 of Section 3.1) -> i.i.d.\\,from (in Latex)

・w.r.t p (in Line 6 of P.6) -> w.r.t.\\,p (in Latex)

・Abbreviation of Brier Score (BS) is declared twice (below Eq.4 and bottom of P.7).

・i+1’th (above Eq.8) -> (i+1)th or (i+1)st

・(SCUT-FBP5500 ?), (Retina-MNIST ?) (in Section 5.1) -> ? (\\cite{} does not work properly.)

・Appendix section A (see A Appendix, B Proof of Lemma 4.1, C Proof of Lemma 4.2 in P.14) is blank.

・Capitalization rule of section titles is not unified (see 4.2 Unimodal Output Probabilities Generation, and G Computation details).

・The following papers are the same and double-counted in References:

-Xintong Shi, Wenzhi Cao, and Sebastian Raschka. Deep neural networks for rank-consistent ordinal regression based on conditional probabilities. arXiv preprint arXiv:2111.08851, 2021.

-Xintong Shi, Wenzhi Cao, and Sebastian Raschka. Deep neural networks for rank-consistent ordinal regression based on conditional probabilities. Pattern Analysis and Applications, 26(3):941–955, 2023.

---

References

R1.
R. Yamasaki, "Unimodal Likelihood Models for Ordinal Data", Transactions on Machine Learning Research.

R2.
R. Yamasaki, T. Tanaka, "Remarks on Loss Function of Threshold Method for Ordinal Regression Problem", arXiv preprint arXiv:2405.13288.

---

> ### Author Response · Authors · 2025-06-25
>
> We thank the reviewer for the thorough and constructive feedback, which has significantly strengthened the quality of our work.
>
> **W1**: We acknowledge the reviewer for the given reference; we changed the method’s name from UNICORNN to UnicornNet.
>
>
> **W2**: We agree that the term "inherent order of classes" was not sufficiently defined in the original submission, which could lead to ambiguity. In response, we have revised paragraph 2 of the introduction to clarify this concept and explain its relevance to ordinal regression.
>
> **W3**:  We sincerely thank the reviewer for highlighting these points, as addressing them has helped strengthen the motivation, findings, and overall contribution of our work.
> We address two key topics in our response: (1) **Cost sensitivity** in the context of ordinal regression, and (2) The choice of an appropriate **decision rule**.
>
> First, we would like to mention that cost sensitivity in ordinal regression is an important topic, and we have revised Section 1 accordingly. In our view, cost sensitivity is essential in ordinal regression, particularly when minimizing MAE, the most widely used metric. This is because MAE relies on the model’s ability to utilize the supervisory signal, which reflects the order and relative distances between classes. Ignoring this signal undermines performance. Moreover, common approaches such as SORD, DLDL, ORCU, and the method by Beckham and Pal incorporate cost-sensitive losses, further highlighting the importance of this perspective.
> While it is true that cost sensitivity may seem less critical when optimizing for accuracy, where all errors are treated equally, we argue that it can still improve performance. To support this, we conducted an experiment comparing two variants of UnicornNet; one trained with a cross-entropy, and the other one using OT. The experiment was done on the RetinaMNIST dataset. Even when accuracy was the evaluation metric, the OT-based model outperformed the one trained with cross-entropy, reinforcing the benefit of incorporating cost-sensitive learning.
> | Loss           | MAE        | ACC        |
> |----------------|------------|------------|
> | Cross-entropy  | 0.87 ± 0.1       | 0.5 ± 0.02       |
> | OT             | **0.67 ±  0.01** | **0.53 ± 0.01** |
>
> Regarding the decision rule, we agree that the optimal decision rule for minimizing MAE is the median predictor: $\arg\min_y \sum_{i=1}^k p(Y=i,|,X=x) |y - i|$. However, similar to many existing methods that incorporate cost-sensitivity in the loss, such as DLDL and ORCU, we adopt the mode predictor $\arg\max_y p(Y = y,|,X = x)$. To support this choice theoretically and to further justify the use of OT as a loss function, we refer the reviewer to the revised Section 3.3, where Proposition 5.2 from \cite{frogner2015learning} is discussed. This result shows that when optimizing OT loss, the expected MAE of the empirical risk minimizer using the mode predictor is provably bounded. We also empirically evaluated the suggested median-based decision rule, but it did not outperform our current mode-based approach. We appreciate the reviewer’s suggestion and agree that aligning the loss with the decision rule is a critical aspect of ordinal regression. This allowed us to further strengthen our theoretical justification.
> | Dataset | Method       | MAE |
> |---------|--------------|-----------------------|
> | FGNET   | Median Predictor    | 0.33 ± 0.09           |
> |         | Mode Predictor| **0.31 ± 0.08**                     |
> | AAF     | Median Predictor    | **0.42 ± 0.01**           |
> |         | Mode Predictor | **0.42 ± 0.01**                     |
> | EVA     | Median Predictor    | 0.57 ± 0.02           |
> |         | Mode Predictor | **0.57 ± 0.01**                     |
>
> **W4**: We kindly refer the reviewer to the end of Paragraph 4 in the Related Work section, where we explicitly cite these references. We also agree with the reviewer that these references should be cited in the introduction section as well. Regarding the theoretical justification for Optimal Transport as the training loss, as pointed out above (W4), we’ve modified Sec. 3.3.
>
>
> **W5**: It is true that, given the reviewer’s example, the MLE estimator will indeed consider not only the predicted probability for the true class, however our intention is that per-sample the MLE ignores the remaining mass and considering only the mass on one class, and that for MLE to evaluate the mass of all of the classes, a sample would needed to appear with all of the classes in the datasets, which practically, does not occur. Again, this discussion falls into the cost-sensitivity of an ordinal regression model, our intention, and many other methods (Beckham and Pal, DLDL, SORD, etc), is that an ordinal regression model needs to be cost-sensitive in order for it not to miss meaningful information such as the order of the labels, and for reducing MAE.

---

> > ### Author Response · Authors · 2025-06-25
> >
> > **W6**: Thank you for providing these references. We have now included and cited them accordingly. There are indeed solutions for multi-label approaches that are consistent, like the one that the reviewer suggested, and others that are already cited (CORN); however, in our provided references, not all of the multi-label approaches are consistent, and the consistent ones are not unimodal.
> >
> > **W7**: The VS-SL method indeed addresses the limited expressiveness of the approach proposed by Beckham and Pal, and we have cited it accordingly. That said, our method is inspired by Beckham and Pal’s work and follows a parametric design. While parametric models may be less expressive than non-parametric ones, they offer benefits such as requiring less data for training. Moreover, our experimental results demonstrate that although our method is less expressive than certain non-parametric unimodal models, such as UnimodalNet, which also employs maximum likelihood estimation (MLE) like VS-SL, our approach consistently outperforms them across all datasets and metrics.
> >
> >
> > **W8**: We would like to clarify that while it is true that the output distribution of POM is not guaranteed to be unimodal for a general function $F$, our claim specifically refers to a particular case where $F$ is the sigmoid CDF (see Sec. 3, Paragraph 1). In this setting, with fixed-length equal-width bins, POM is indeed guaranteed to produce a unimodal distribution.
> >
> >
> > **W9**: We thank the reviewer for his suggestion, we’ve added a new subsection 5.3 regarding the experimental setup of the different experiments.
> >
> > **W10**: Due to the large number of datasets, model variants, and the computational cost of the training, we limited each setting to five trials. This reflects a practical trade-off given time and resource constraints, and is in line with common practice.
> >
> > **W11+W12**: We rephrased the relevant sentences to better align with the reviewer’s comments.
> >
> >
> >
> > **W13**: We have added brief explanations of key approaches, including the use of soft targets in prior works, to improve clarity. However, we did not detail every method to avoid overloading the reader. Instead, we focused on highlighting common limitations to better motivate the contribution of UnicornNet.
> >
> >
> > **W14**: We want to clarify that the need for calibration in UnicornNet is not due to the truncated normal distribution, but rather stems from the use of optimal transport loss, as discussed in our paper. However, as shown in Table 1, other methods are also uncalibrated. For example, DLDL and SORD suffer from miscalibration as a result of their usage of predefined target distributions that do not reflect uncertainty, or Beckham, which also uses OT. This is not a specific limitation of UnicornNet; rather, UnicornNet is the only method that actively addresses this issue.
> >
> > **W15**: Our method indeed consists of three components: unimodality, OT loss, and calibration.
> > Unimodality is a design principle, not a removable module—it ensures consistency with the ordinal label structure, and removing it would contradict the goal of ordinal regression. Therefore, it is not ablated.
> > For calibration, as mentioned, we provided an ablation in the paper. For the loss, as suggested, we added a further comparison using Cross-Entropy loss instead of OT on RetinaMNIST (Section 5.8.2). The results show OT significantly outperforms CE, highlighting its importance.
> > Regarding the suggestion to include models such as Beckham & Pal or R1 in the ablation, we believe this is less relevant for an ablation of our method. These models differ not only in their likelihood formulation but would also require changes to the calibration process, making direct ablation difficult and less informative for isolating components of our approach. We do reference and compare Backham and Pal in the experimental section to provide appropriate context.

---

> > > ### Author Response · Authors · 2025-06-25
> > >
> > > **W16**: We have added the needed citation for each metric used.
> > >
> > > **W17**: We have revised the tables to ensure that all reported values use consistent significant digits.
> > >
> > > **W18**:  We thank the reviewer for their detailed and helpful feedback. We have carefully revised the manuscript to address all the concerns:
> > > * Correction of Eq. (1): We fixed the mathematical inconsistency where the left-hand side implied a conditional probability while the middle term implied a joint distribution. We rewrote the expression to reflect a consistent and correct probabilistic interpretation.
> > >
> > > * Removal of unnecessary notation: We removed the introduction of superfluous terms above Eq. (1) and within Eq. (2), as suggested.
> > >
> > > * Brackets unification: We have unified the notation of brackets throughout the equations, ensuring consistency between Eq. (1), Eq. (3), and the rest of the paper.
> > >
> > > * Clarification of notation: We defined $Y$ on Line 9 on page 2.
> > >
> > > * Vector notation: We have revised all vector-valued variables to use consistent bold notation (\bm{}), following the reviewer’s suggestion to clearly distinguish them from scalars.
> > >
> > > * Equation simplification: Eq. (2) and Eq. (7) were rewritten in a more compact and readable one-line format.
> > >
> > > * Wording correction: The phrase "given the unknown real distribution Y∣X=x" has been rephrased to “given the unknown true conditional distribution of $Y$ given $X=x$”.
> > >
> > > * Marginal vs. conditional clarity: We revised instances where marginal and conditional probabilities were conflated, ensuring that all probability terms are now explicitly and correctly labeled.
> > > * Calibration definition: We added a citation to the suggested reference Wang (2023), We rewrote the expression in a way that is consistent with our paper’s notation.
> > >
> > > **W19**:  Thank you for highlighting these writing and formatting issues. We have made the following corrections:
> > > * Replaced straight quotes (e.g., "1") with proper LaTeX quotes (``1'').
> > > * Fixed spacing in abbreviations (e.g., i.i.d.\,from, w.r.t.\,p).
> > > * Removed the duplicate definition of the Brier Score (BS).
> > > * Replaced i+1’th with (i+1)th.
> > > * Fixed broken citations (e.g., SCUT-FBP5500, Retina-MNIST).
> > > * Populated Appendix sections A–C with the appropriate content.
> > > * Unified capitalization in section titles.
> > > * Removed the duplicated reference.
> > >
> > >
> > > [frogner2015learning] https://arxiv.org/abs/1506.05439

---

> > > > ### Comment · Reviewer_j8ZT · 2025-06-26
> > > > **2nd review 1**
> > > >
> > > > I have responded hastily, considering the deadline for the review.
> > > > If you find it difficult to understand the meaning of my response, please ask additional questions.
> > > >
> > > > My major concerns are
> > > > - The proposal of a unimodal likelihood model based on a truncated Gaussian is not properly motivated, and there is insufficient comparison and discussion with other similar methods (W7,15).
> > > > - I disagree with the criticism of MLE (W5).
> > > > - The decision rules of classifiers learned with MLE, such as POM, can be inappropriate (for MAE), making the experimental comparison unfair (W2,3,15).
> > > > - Others (W6,8,18).
> > > >
> > > > ---
> > > >
> > > > W1.
> > > > OK
> > > >
> > > > W2.
> > > > The added sentence "A straightforward way to leverage the label order is to use a cost-sensitive loss function — one in which the penalty for misclassification increases strictly with the ordinal distance between the predicted and true labels." is misleading. When we would like to minimize miss-classification rate for ordinal data, using a cost-sensitive loss function yields a statistically inconsistent classifier. How is the use of cost-sensitive losses statistically/theoretically advisable in ordinal regression? (continue to W3)
> > > >
> > > > W3.
> > > > Regarding the cross-entropy and MAE in the first table of the authors' response, did the authors use cross-entropy for learning, use a median predictor as the decision rule, and then evaluate MAE?
> > > > Judging from Table 1 in the revised paper, it appears that my points were not reflected.
> > > > For example, when you trained the POM by cross entropy, then you should use the median predictor when evaluating MAE (this is the cost-sensitive convention).
> > > >
> > > > I pointed out "In cost-sensitive learning, it is standard to change either the learning objective function or the decision function. This study appears to have adopted the former strategy, but did not adequately address the latter strategy. Perform experiments with the latter strategy as well."
> > > > The authors added theoretical support for the former strategy (Proposition 5.2).
> > > > This is good (but I am not saying that you should change your decision function when learning with OT).
> > > > However, the authors considered a combination of cross entropy and mode predictor only, and the latter strategy is not appropriately performed.
> > > > This is unfair.
> > > >
> > > > W4.
> > > > OK

---

> > > > > ### Comment · Reviewer_j8ZT · 2025-06-26
> > > > > **2nd review 2 (fin)**
> > > > >
> > > > > W5.
> > > > > The sentence "as it only considers the probability mass the model assigns to the true class, ignoring the remaining mass" does not deliver the authors' intention.
> > > > > I think it should be rewritten.
> > > > >
> > > > > However, even though the sentence will be rewritten, I do not agree with the authors' intention (since Cramer-Rao justifies MLE even for ordinal data).
> > > > > I will leave the matter to the other reviewers to make a judgment.
> > > > >
> > > > > W6.
> > > > > It is true that multi-label approaches are not necessarily unimodal.
> > > > > However, even if the multi-label approaches were not consistent when they were proposed, that problem has now been solved.
> > > > > Nevertheless, pointing out that problem would be harmful to readers (though it might be helpful to write that the problem has been solved).
> > > > > If readers of this paper set out to solve that problem, how will the authors take responsibility for wasting their time?
> > > > >
> > > > > I am not encouraging authors to cite R1 and R2.
> > > > > I am merely pointing out that at least in R1 and R2 the problems with multi-label approaches have been addressed.
> > > > > I hope that the authors will look for more appropriate references.
> > > > >
> > > > > W7.
> > > > > In this response, I understood that the authors would like to use a unimodal likelihood model that has higher representation ability than (Beckham & Pal, 2017) and lower representation ability than VS-SL model (this should be written in the paper, but not).
> > > > > The previous work R1 proposed PO-VS-SL, PO-ORD-ACL, OH-VS-SL, OH-ORD-ACL, OH-BIN, and OH-POI models as unimodal likelihood models that have higher representation ability than (Beckham & Pal, 2017) and lower representation ability than VS-SL model.
> > > > > The paper only describes (Beckham & Pal, 2017) and POM to motivate truncated Gaussians, and its proposal is not adequately motivated.
> > > > > What is the usefulness of the truncated Gaussian compared to R1’s various models?
> > > > > The authors should discuss the peculiarities of truncated Gaussians more carefully in the paper.
> > > > >
> > > > > W8.
> > > > > The response is not correct.
> > > > > In the case $k=3$, $(F(0-0),F(1-0)-F(0-0),1-F(1-0))\\approx(0.5,0.23,0.27)$ ($\alpha=(0,1)$).
> > > > > Such a case exists even when $k\\neq3$.
> > > > >
> > > > > W9.
> > > > > OK
> > > > >
> > > > > W10.
> > > > > OK
> > > > >
> > > > > W11.
> > > > > OK
> > > > >
> > > > > W12.
> > > > > OK
> > > > >
> > > > > W13.
> > > > > See W6 and W7.
> > > > >
> > > > > W14,15.
> > > > > I understand the authors’ argument regarding the need for calibration.
> > > > > I would like to know the impact that calibration or not has on classification performance (it is natural that calibration would improve ECE).
> > > > >
> > > > > I apologize for misunderstanding the meaning of ablation study.
> > > > > In addition to the above things about calibration, I recommended to
> > > > > - compare a proposed unimodal likelihood model and other unimodal likelihood models in (Beckham & Pal, 2017) and the previous works R1 (some of PO-VS-SL, PO-ORD-ACL, OH-VS-SL, OH-ORD-ACL, OH-BIN, and OH-POI models) while keeping the training objective function the same,
> > > > > - compare OT minimization and maximum likelihood method for the proposed truncated Gauss likelihood model.
> > > > >
> > > > > The first recommendation is related to W7 (to show the necessity and significance of Gaussian truncation), and this is not on this paper.
> > > > > The second recommendation is related to W3, and this is on the paper, but I am concerned about its fairness (write experimental details).
> > > > >
> > > > > W16.
> > > > > OK
> > > > >
> > > > > W17.
> > > > > OK
> > > > >
> > > > > W18.
> > > > > Marginal vs. conditional clarity: Sec.1, Para.3, P.2; Sec.2, Para.1, P.2.
> > > > > Insufficient.
> > > > >
> > > > > W19.
> > > > > OK

---

> > > > > > ### Author Response · Authors · 2025-06-27
> > > > > >
> > > > > > We appreciate the reviewer’s thorough and thoughtful feedback.
> > > > > >
> > > > > > **W2:** First, we revised the sentence to: “A straightforward way to leverage the label order, specifically when the evaluation metric is ordinal such as Mean Absolute Error (MAE), is to use a cost-sensitive loss function in which the penalty for misclassification increases strictly with the ordinal distance between the predicted and true labels.”
> > > > > >
> > > > > > Second, since MAE is the most commonly used evaluation metric in ordinal regression, rather than accuracy, adopting a cost-sensitive loss function is both appropriate and desirable.
> > > > > >
> > > > > > **W3:** The study focuses on modifying the learning objective by incorporating a loss function that accounts for ordinal structure, as supported by the theoretical results in Proposition 5.2. This direction was chosen deliberately to examine how the choice of loss function alone influences performance, while keeping the decision rule fixed across all methods. Specifically, the mode predictor was used consistently to allow a controlled comparison of different training objectives.
> > > > > > Although using the median predictor may be more appropriate when evaluating with MAE under cross-entropy training, we chose not to adjust the decision rule in order to avoid confounding the effects of the loss and the inference strategy. This decision was made to ensure fairness and clarity in interpreting the role of the training objective.
> > > > > >
> > > > > > It is also important to note that the evaluation in the paper is not restricted to MAE. Multiple performance metrics, including OOA and others, are reported to capture a comprehensive view of model behavior. This broader perspective supports a more balanced assessment beyond a single evaluation criterion.
> > > > > > We acknowledge that incorporating alternative decision rules such as the median could provide additional insights and will highlight this as a limitation of the current study.
> > > > > >
> > > > > > **W5:** We rephrased the sentence to: “We argue that maximum likelihood is not a suitable measure of quality in the context of ordinal regression, as per sample, it only considers the probability mass the model assigns to the true class, ignoring the remaining mass, and behaves as a non-cost-sensitive loss. This implicitly assumes that ``all mistakes are equal'', which, as discussed above, is not ideal for minimizing order-aware evaluation metrics in ordinal regression.”
> > > > > >
> > > > > > **W6**: We have already stated that the inconsistency problem is already solved by these methods. However, the key limitation we highlight is their lack of unimodality. “Second, *even if the output probabilities are consistent, as is the case in Liu et al. (2018a); Shi et al. (2023); Cao et al. (2020)*, the predicted class probabilities are not necessarily unimodal.”
> > > > > >
> > > > > > **W7:** The key advantage of the truncated Gaussian mechanism used in UnicornNet over the different R1’s methods is its ability to decouple the location and shape of the output distribution via two learned parameters (μ, σ). This enables us to perform an accuracy-preserving calibration by varying only the scale parameter.
> > > > > >
> > > > > > **W8:** We are sorry for this incorrectness. We have rephrased the sentence to: “the lack of unimodality of POM can be fixed by modifying the CDF, letting the bins be of equal length and remain fixed during training.”
> > > > > >
> > > > > > **W14,15:** Regarding the impact of calibration on classification performance, our calibration method is accuracy-preserving. Meaning, under the mode predictor, it retains the same class predictor, and thus does not affect classification accuracy. The effect of this calibration is reflected in improved calibration metrics such as ECE, without modifying the classification performance.
> > > > > >
> > > > > > While we did not include the specific unimodal likelihood models from R1, our experiments compare the proposed truncated Gaussian model with a diverse set of SOTA competitive baselines, including highly expressive unimodal models such as UnimodalNet. Furthermore, Beckham & Pal (2017) already utilize an OT-based loss, and our results in Table 1, using the same OT loss for UnicornNet, demonstrate significant improvements, highlighting the benefit of our likelihood formulation.
> > > > > >
> > > > > > Our intention in this work is to evaluate the proposed model under a consistent training objective against already established models. Extending prior methods (e.g., R1 models) with losses they were not originally designed with, such as OT, would involve substantial methodological changes. As such, we consider those comparisons to be outside the scope of this paper.
> > > > > >
> > > > > > **W18:** In the first point, we present the general definition of a unimodal distribution, which is not specific to the model's output conditional probability distribution. Regarding the second point, we apologize for the issue; it has been corrected.

---

> > > > > > > ### Comment · Reviewer_j8ZT · 2025-06-27
> > > > > > > **3rd review**
> > > > > > >
> > > > > > > My major concerns (2nd review 1) are not resolved.
> > > > > > > The authors do not appear to be willing to address my concerns.
> > > > > > > I will leave them to other reviewers to judge.
> > > > > > >
> > > > > > > ---
> > > > > > >
> > > > > > > *W2:
> > > > > > > The revised sentence is not appropriate as a sentence characterizing ordinal regression.
> > > > > > > If you want to minimize a cost-sensitive loss (e.g. MAE) regardless of the order of the data, then you can use that cost-sensitive loss (e.g. MAE) for learning.
> > > > > > > This is a convention of the cost-sensitive learning that has nothing to do with ordinal regression.
> > > > > > >
> > > > > > > The choice of objective function and cost is up to the user.
> > > > > > > In many cases, one wants to minimize the misclassification rate as well even for ordinal data.
> > > > > > > It is not a good method to always minimize the MAE while ignoring the cost the user wants to use.
> > > > > > >
> > > > > > > *W3:
> > > > > > > I think that's unfair.
> > > > > > >
> > > > > > > To begin with, I am against evaluating a single prediction using multiple criteria.
> > > > > > > If we really want to evaluate using multiple criteria, it will be better to output the results according to a multi-objective optimization approach.
> > > > > > >
> > > > > > > Rather, it would be fairer and more persuasive to use only MAE (and a few other criteria) for which a theoretically appropriate comparison is possible, and to compare methods using settings (e.g., decision rules) optimized for each evaluation criterion.
> > > > > > >
> > > > > > > At least, not optimizing the decision rule for e.g. POM under-reports the capabilities of the POM.
> > > > > > > Not only the comparison in this study is insufficient to demonstrate the effectiveness of the proposed method, but also this is disrespectful to previous works.
> > > > > > >
> > > > > > > *W5:
> > > > > > > Again, I do not agree with the authors' intention (since Cramer-Rao justifies MLE even for ordinal data).
> > > > > > > I will leave the matter to the other reviewers to make a judgment.
> > > > > > > See also W2 of 3nd review.
> > > > > > >
> > > > > > > W6:
> > > > > > > OK
> > > > > > >
> > > > > > > *W7:
> > > > > > > The authors' response is incorrect.
> > > > > > > At least, in the OH-BIN and OH-POI models of R1, the conditional modes does not change even if the scale-related parameters are changed.
> > > > > > >
> > > > > > > W8:
> > > > > > > Ok.
> > > > > > > But, the flow of the paragraph including that modified sentence is unnatural.
> > > > > > >
> > > > > > > *W14,15:
> > > > > > > (former)
> > > > > > > OK.
> > > > > > > I had considered that the accuracy-preserving calibration does not change accuracy but should change MAE, and other evaluation metrics, but I understand now that it does not change MAE, and other evaluation metrics as well since the decision rule is a mode-predictor.
> > > > > > >
> > > > > > > (latter)
> > > > > > > At least, in the OH-BIN and OH-POI models of R1, which model a scale-parameter of Beckham & Pal (2017) with a function, the conditional modes does not change even if the scale-related parameters are changed.
> > > > > > > Therefore, we can apply accuracy-preserving calibration to OH-BIN and OH-POI models and learn those models with OT loss.
> > > > > > > If the authors would like to use a unimodal likelihood model that has higher representation ability than (Beckham & Pal, 2017) and lower representation ability than VS-SL model, it will be natural to apply OH-BIN and OH-POI models in present.
> > > > > > > Proposing truncated Gaussian without any discussion is not a constructive development.
> > > > > > > The paper should describe why it proposes truncated Gaussian as one of unimodal likelihood models and the proposed method should actively adopt that model.
> > > > > > > The authors are free to leave such a comparison outside the scope of their paper, but I think that such a paper is poorly argued.
> > > > > > >
> > > > > > > W18:
> > > > > > > OK.
> > > > > > > But, to avoid misunderstanding among readers, I suggest changing the first sentence to "A distribution of a random variable $Y\\in\\{1,\\ldots,k\\}$ is called unimodal if there exists $j\\in\\{1,\\ldots,k\\}$ such that $\\mathbb{P}(Y=1)\\le\\cdots\\le \\mathbb{P}(Y=j)\\ge\\cdots\\ge\\mathbb{P}(Y=k)$."
> > > > > > >
> > > > > > > W20:
> > > > > > > Please refrain from making mistakes in your revised text.
> > > > > > > - "Wu et al. (2023)) However" (in Sec.1, Para.2, P.1) loses a comma.
> > > > > > > - " of either "1" or "4", but not "2" or "3" (in Sec.1, Para.3, P.2)

---

> > > > > > > > ### Comment · Action_Editor_zBmQ · 2025-06-27
> > > > > > > >
> > > > > > > > Thanks to the authors and the reviewer for the interactive discussions.
> > > > > > > >
> > > > > > > > As my name (AE) is public, I believe that there is no conflict interest in enriching the discussion of W3 by pointing everyone to my own work
> > > > > > > >
> > > > > > > > https://www.csie.ntu.edu.tw/~htlin/paper/doc/redordinal.pdf
> > > > > > > >
> > > > > > > > given that I have done extensive research on cost-sensitive learning and ordinal regression (PhD thesis) in my early career. Look forward to hearing everyone's professional thoughts on the connections.

---

### Review · Reviewer_zcLd · 2025-06-08

**Summary Of Contributions:**

The paper proposes UNICORNN, a deep ordinal-regression framework that (1) architecturally guarantees unimodal class–probability outputs by slicing a truncated normal density into equal-width bins, (2) optimises an Optimal-Transport (OT) loss to respect class order, and (3) adds an “accuracy-preserving” calibration phase that re-estimates only the scale parameter via Brier loss. Experiments on six image benchmarks show state-of-the-art MAE and markedly lower Expected Calibration Error (ECE) relative to recent unimodal-ordinal baselines such as DLDL, SORD and UnimodalNet. Ablations confirm the necessity of the post-hoc calibration for ECE.

**Audience:**

Yes

**Broader Impact Concerns:**

N/A.

**Claims And Evidence:**

Yes

**Requested Changes:**

Add the related work [1-6] to discussion and comparison, to acknowledge previous work on regression ordinality and demonstrate the performance in imbalancing cases.

Add datasets in non-imaging domains.

**Strengths And Weaknesses:**

Strengths:

Paper is well written; algorithms specified; appendix gives training details.

The experimental evaluation is comprehensive, including six diverse, real-world datasets, with the use of multiple, standard evaluation metrics.

Elegant two-parameter construction that provably yields unimodal probabilities. Highlights a clear OT-versus-calibration trade-off seldom discussed in ordinal work

Weakness:

There are previous work on the ordinality of regression missing in the discussion/comparison, for example, [1] and [2]. Furthermore, many ordinal problems are also highly imbalanced regression tasks, the paper would benefit from acknowledging that link in the related-work section and testing UNICORNN under class-skewed regimes. There is a series of previous work on imbalanced regression including [3], [4], [5], and [6].

The model requires a two-phase training procedure, which introduces additional complexity and computational overhead compared to end-to-end trained models. While the calibration phase only tunes a subset of parameters, it still necessitates a second full pass through the training data.

While the truncated normal distribution is a reasonable choice, a brief justification for why this specific distribution was chosen over other two-parameter unimodal distributions (e.g., Beta distribution on the [-1, 1] interval) would be beneficial.

While Lemma 4.2 shows that the argmax (the predicted class) is preserved, the term "accuracy-preserving" could be interpreted more broadly. The calibration step will change the full probability vector, which could affect metrics that depend on it (like QWK). A clarification on this point would be useful. The MAE is also shown to improve, which contradicts a strict interpretation of "accuracy-preserving."

All datasets are image based; no evidence on tabular, NLP or time-series—hurts claim of “many real-world applications”. Suggest to add the datasets used in [1-6].

[1] Rank-N-Contrast: Learning Continuous Representations for Regression. NeurIPS 2023.

[2] Mixup Your Own Pairs. Arxiv 2023.

[3] Delving into Deep Imbalanced Regression. ICML 2021.

[4] Ranksim: Ranking similarity regularization for deep imbalanced regression. ICML 2022.

[5] Conr: Contrastive regularizer for deep imbalanced regression. ICLR 2023.

[6] Improve Representation for Imbalanced Regression through Geometric Constraints. CVPR 2025.

---

> ### Author Response · Authors · 2025-06-25
>
> We appreciate the reviewer’s recognition of our contributions and are grateful for the valuable feedback provided.
>
>
> **W1 + C1**: We thank the reviewer for highlighting these relevant works. We agree that [1] and [2] address important aspects of ordinality in regression, and we have added them to the first paragraph of the related work section to acknowledge their contributions properly.
> Regarding the works on imbalanced regression [3–6], while they are indeed valuable in handling class-skewed scenarios, they fall outside the direct scope of our current study. Our focus in this work is on evaluating UNICORNN within the standard ordinal regression setting using widely adopted benchmark datasets. That said, exploring how UNICORNN performs under imbalanced regimes and potentially adapting it to such settings is an interesting direction for future work.
>
>
> **C2**: Following the reviewer’s suggestion, we conducted an additional experiment on a non-image dataset: the balanced Fireman tabular dataset, using the same setup as introduced by CORN. We also extended our benchmark to include two state-of-the-art ordinal regression methods: CORN, a rank-consistent approach, and ORCU, which is specifically designed to produce calibrated probability estimates. The updated results are presented in the revised Table 1. As shown, UNICORNN achieves the lowest MAE across all datasets except Adience and Fireman, where the difference from the first place is negligible. In terms of calibration, UNICORNN consistently provides well-calibrated predictions, achieving the lowest ECE in most cases and showing only negligible differences otherwise.
>
>
> **W2**:  While UNICORNN involves a two-phase training procedure, we would like to emphasize that the calibration phase is lightweight and efficient. It optimizes only a very small subset of parameters, $\theta_{\sigma}$, which introduces minimal computational overhead compared to the initial training of the full model. To demonstrate it, we have added a new section (Section 5.6) analyzing the runtime of each phase. This analysis demonstrates that the calibration step contributes only marginally to the overall training time, supporting the practicality of our approach. Moreover, while it is true that the calibration phase still necessitates a second full pass through the training data, we can forward the entire training data through the backbone as a preprocessing step and train the $\theta_{\sigma}$ parameters on top of these transformed samples.
>
> **W3**: In line with the reviewer's suggestion, we have added an explanation for the choice of the truncated normal distribution in Sec. 4.3:
> “Intuitively, to preserve accuracy, the distribution should be unimodal and symmetric around its mode. This motivates our choice of the truncated normal distribution.”
>
> **W4**: We have added a clarification for it in Sec 4.3:
> “although the probability output distribution is indeed modified”.

---

### Review · Reviewer_XFgA · 2025-06-14

**Summary Of Contributions:**

This paper introduces UNICORNN (*UNImodal Calibrated Ordinal Regression Neural Network*), a deep learning approach for ordinal regression that addresses three key requirements: (1) ensuring unimodal output probabilities through architectural design, (2) capturing ordinal relationships using optimal transport (OT) loss, and (3) providing well-calibrated probability estimates via post-training calibration.

The main contributions are:

- The authors identify that optimal transport loss tends to favor peaked distributions over calibrated ones in ordinal regression, which appears to be a new observation in the literature.

- UNICORNN uses truncated normal distributions with fixed equal-length bins to guarantee unimodal outputs, controlled by learnable location (μ) and scale (σ) parameters.

- The method first trains both μ and σ mappings using OT loss, then performs accuracy-preserving calibration by retraining only the σ mapping using Brier Score while keeping μ fixed.

- Formal proofs are provided for unimodality (Lemma 4.1; proof provided in Appendix B) and accuracy preservation during calibration (Lemma 4.2; proof provided in Appendix C).

- Experiments on six image datasets demonstrate competitive performance while guaranteeing unimodality.

**Audience:**

Yes

**Broader Impact Concerns:**

The paper does not include a Broader Impact Statement, which should be added given the potential applications in sensitive domains like medical diagnosis and credit rating mentioned in the introduction. Key concerns to address:

1. When used for medical severity grading or diagnostic tasks, miscalibrated probabilities could lead to inappropriate treatment decisions.

2. Ordinal regression models in domains like credit rating or hiring could perpetuate or amplify existing biases if not carefully validated across different demographic groups.

3. While the method aims to improve calibration, the implications of remaining miscalibration in high-stakes applications should be discussed.

**Claims And Evidence:**

Yes

**Requested Changes:**

**Critical for acceptance:**

1. Include comparison with recent methods like CORN on additional datasets beyond the appendix. The current CORN comparison is limited and should be expanded.

2. Include additional experiments to improve the experimental evaluation section.

3. Include non-image datasets and comparison with recent calibration-focused methods like Oridnal Regression loss for Calibration and Unimodality (ORCU).

4. Provide analysis of sensitivity to key hyperparameters for each training phase.

5. Include computational complexity analysis and runtime comparisons with baseline methods.


### **Would strengthen the work:**

1. Add calibration plots or reliability diagrams to better visualize the calibration improvements claimed.

2. Beyond the calibration phase, analyze other components like the effect of different truncation bounds or bin numbers.

3. Provide more detailed algorithmic descriptions and clearer guidance on hyperparameter selection.s

**Strengths And Weaknesses:**

**Strengths:**

1. The paper clearly identifies limitations in existing ordinal regression methods and articulates three desirable properties that current approaches fail to achieve simultaneously.

2. The identification that OT loss prioritizes peaked over calibrated distributions (Section 3.3) is valuable and provides important understanding of OT's behavior in ordinal regression.

3. The truncated normal approach with fixed bins effectively addresses POM's unimodality issues while providing more flexibility than single-parameter methods.

4. Lemmas 4.1 and 4.2 provide formal guarantees for unimodality and accuracy preservation, respectively.

5. UNICORNN achieves competitive or superior results across datasets.


**Weaknesses:**

1. While effective, the truncated normal approach represents an incremental improvement over existing architectural solutions rather than a fundamental breakthrough.

2. The approach requires careful tuning of two separate training phases, which may complicate implementation and hyperparameter selection.

3. Experiments focus primarily on image-based tasks; broader domain evaluation would strengthen generalizability claims.

4. Expanding more on the experimental evaluation would help improve the manuscript.

5. No comparison with recent methods like ORCU that also address calibration in ordinal regression

6. While Lemmas 4.1 and 4.2 are useful, deeper theoretical analysis of the OT-calibration trade-off would strengthen the contribution

7. Some citations are missing:

  - *facial beauty prediction (SCUT-FBP5500 ?),*

  - *bio-medical image classification (Retina-MNIST ?)*

8. No discussion of the computational cost implications of the two-stage training procedure.

---

> ### Author Response · Authors · 2025-06-25
>
> We sincerely thank the reviewer for recognizing the value of our work and for offering such thoughtful and constructive feedback.
>
> **W1**: While we agree with the reviewer that the truncated normal approach represents an incremental improvement rather than a fundamental breakthrough on its own, we emphasize that UNICORNN’s key innovation lies in the integration of all three components. This integration addresses challenges that previous methods not only failed to resolve but also did not explicitly consider.
>
> **W2**: We thank the reviewer for this observation. We would like to clarify that while the first phase of training, which consists the optimization of the full backbone model, does indeed involve extensive training and hyperparameter selection, the second phase is substantially simpler. Specifically, the calibration phase involves optimizing only the $\theta_{\sigma}$ parameter set, which is orders of magnitude smaller in size and complexity. As a result, this second phase is lightweight, both in terms of computation and the need for hyperparameter tuning, and does not significantly complicate the overall implementation.
>
> **W4 + W3 + C2 + C4 + C3**:  In response to the reviewer’s suggestion, we have expanded the experimental evaluation in several meaningful ways:
> * We conducted an additional experiment on a non-image dataset: the balanced Fireman tabular dataset, using the same setup as introduced by CORN. We also extended our benchmark to include two state-of-the-art ordinal regression methods: CORN, a rank-consistent approach, and ORCU, which is specifically designed to produce calibrated probability estimates. The updated results are presented in the revised Table 1. As shown, UNICORNN achieves the lowest MAE across all datasets except Adience and Fireman, where the difference from the first place is negligible. In terms of calibration, UNICORNN consistently provides well-calibrated predictions, achieving the lowest ECE in most cases and showing only negligible differences otherwise.
>
> * To address concerns regarding the computational cost of our two-phase training procedure, we have introduced a new section (Section 5.6) analyzing the runtime of each phase.
>
> * Finally, we have included a hyperparameter sensitivity analysis (Section 5.7), focusing in particular on the cost metric used in the OT loss, demonstrating how different configurations affect performance.
> These additions are now reflected in the revised manuscript.
>
> **W5 + C1 + C3**: As explained above, as suggested by the reviewer, we have included CORN and ORCU in the experimental evaluation across all datasets presented in the manuscript.
>
> **W6**: We thank the reviewer for pointing out the importance of further theoretical analysis. In Section 3.3, we provide an illustrative example that demonstrates the trade-off between OT loss and calibration empirically. While we agree that a deeper theoretical exploration could enrich the understanding of this trade-off, we would like to note that our primary focus in this work is on the practical identification and mitigation of this effect through model design and calibration. We believe that a formal theoretical treatment would require a separate, dedicated analysis and consider it a promising direction for future work. We also want to mention that we have added a new theoretical motivation in Section 3.3 for the usage of OT as a loss function in ordinal regression tasks.
>
> **W7**: We have added the missing citations.
>
> **W8 + C5**: In response to the reviewer’s suggestion, we have added a new section analyzing the computational cost associated with both training phases.
>
>
> **Broader Impact**: We have included a Broader Impact Statement discussing potential societal implications, particularly in high-stakes domains such as healthcare and finance.

---

> > ### Comment · Reviewer_XFgA · 2025-06-27
> > **Some comments for the authors**
> >
> > W2: My primary concern was hyperparameter selection in both phases and how choices in one phase influence performance in the subsequent phase.
> >
> > W4 + W3 + C2 + C4 + C3: The results for ORCU on the Retina-MNIST dataset are missing. Additionally, the runtime comparison should include the proposed method alongside the baselines to provide a clearer understanding of its computational performance relative to existing approaches.

---

### Author Response · Authors · 2025-06-25

We would like to sincerely thank all the reviewers for their thoughtful and constructive feedback. Your comments have significantly contributed to strengthening the quality and clarity of our work. In response, we have prepared a revised version of the paper, in which all changes are highlighted in red for ease of reference. Below are our detailed responses to each reviewer.

---

### Note · Authors · 2025-08-01

**Comment:**

We sincerely thank the action editor and reviewers for their thoughtful feedback. Based on these reviews, we have decided to withdraw the submission in order to revise and improve the work accordingly.

**Withdrawal Confirmation:**

I have read and agree with the venue's withdrawal policy on behalf of myself and my co-authors.